# Of Mice and Machines:
# A Comparison of Learning Between Real World Mice and RL Agents

**Shuo Han** [1]   **German Espinosa** [2]   **Junda Huang** [1]   **Daniel A. Dombeck** [3]   **Malcolm A. MacIver** [2]   **Bradly C. Stadie** [1]

## Abstract

Recent advances in reinforcement learning (RL) have demonstrated impressive capabilities in complex decision-making tasks. This progress raises a natural question: how do these artificial systems compare to biological agents, which have been shaped by millions of years of evolution? To help answer this question, we undertake a comparative study of biological mice and RL agents in a predator-avoidance maze environment. Through this analysis, we identify a striking disparity: RL agents consistently demonstrate a lack of self-preservation instinct, readily risking "death" for marginal efficiency gains. These risk-taking strategies are in contrast to biological agents, which exhibit sophisticated risk-assessment and avoidance behaviors. Towards bridging this gap between the biological and artificial, we propose two novel mechanisms that encourage more naturalistic risk-avoidance behaviors in RL agents. Our approach leads to the emergence of naturalistic behaviors, including strategic environment assessment, cautious path planning, and predator avoidance patterns that closely mirror those observed in biological systems.

## 1. Introduction

Mathematical foundations for reinforcement learning (RL) emerged through several landmark contributions: Samuel's early work on learning systems, Bellman's dynamic programming principles, and later Sutton's temporal difference learning (Samuel, 1959; Bellman, 1966; Sutton, 1988). While these seminal advances have enabled powerful decision-making agents, they arose primarily from computational principles[1] rather than biological experimentation. Although subsequent research has attempted to draw parallels between RL algorithms and biological learning mechanisms (Pozzi et al., 2018; Neftci & Averbeck, 2019; Tan et al., 2023), suggesting the biological plausibility of methods like Q-learning, a fundamental question remains: To what extent do RL agents truly capture the decision-making characteristics of biological organisms?

To investigate this question, we designed a controlled experimental environment where mice navigate a complex space while avoiding a robotic predator-like autonomous agent (hereafter referred to as "predator" for brevity). A simulation environment replicates this with the prey controlled by an RL algorithm. The predator is a reactive agent in both the experiment and simulation. Through careful behavioral analysis, offline RL modeling (Levine et al., 2020) of mouse behavior, and Exploratory Data Analysis over visitation density graphs, we identified several striking disparities between biological and artificial agents. Most notably, RL agents demonstrate a remarkable lack of self-preservation instinct, often choosing marginally more efficient paths that bring them dangerously close to the predator. In contrast, biological agents exhibit sophisticated risk-assessment behaviors, with mice spending a significant amount of time gathering environmental information and evaluating predator positions before movement.

Inspired by these observations, we developed two novel mechanisms to bridge the behavioral gap. First, we introduce a modified replay buffer mechanism that amplifies and frequently resamples near-death experiences during training, mimicking post-traumatic stress responses observed in biological systems. This enhancement enables agents to develop more appropriate risk-avoidance behaviors. Second, we propose a fundamental modification to the TD-learning framework incorporating action uncertainty through Q-value variance estimation. These additions create naturally risk-averse agents that better reflect the cautious decision-making

[1]Department of Statistics and Data Science, Northwestern University, Evanston, USA [2]Department of Mechanical Engineering, Northwestern University, Evanston, USA [3]Department of Neurology, Northwestern University, Evanston, USA. Correspondence to: Shuo Han <TansioHan@u.northwestern.edu>.

*Proceedings of the 42ⁿᵈ International Conference on Machine Learning*, Vancouver, Canada. PMLR 267, 2025. Copyright 2025 by the author(s).

---

[1]While Sutton et al. (1998) has some biological inspiration, there is a clear distinction to be drawn between biologically inspired and derived from measured biological data. We investigate the latter viewpoint in this volume.

patterns observed in biological systems. Empirically, these mechanisms enhance the behavioral similarity between artificial and biological agents, increasing the visitation pattern overlap with mice from 20.9% to 86.1%. The enhanced agents exhibit conservative movement patterns with approximately 45% less distance traveled during initial environment entry, closely matching the cautious behavior of real mice.

Given the fundamental behavioral differences we observed between RL and biological agents, we were left wondering whether more advanced AI systems might naturally exhibit more biologically-aligned behavior. To investigate this, we introduced a third agent derived from a large language model (LLM)—a system trained on vast amounts of human-written text and capable of sophisticated world modeling. (Devlin, 2018; Brown et al., 2020; Hurst et al., 2024) Surprisingly, when controlling a simulated version of the mouse, the LLM exhibited risk-taking characteristics much more aligned with traditional RL agents than biological ones. This finding suggests that biological alignment in decision-making behaviors may be a broader challenge in AI systems, independent of their training approach or world knowledge.

Our key contributions can be stated as follows:

- We perform a systematic analysis of behavioral differences between biological and artificial agents in a high-risk pseudo-predator-prey environment.

- We derive novel mechanisms for incorporating biologically inspired risk assessment and experience processing into RL objectives.

- Results show that incorporating these mechanisms lead to trajectories that are closer to biological behavior. This is true even when compared to other surprise-avoiding objectives from the literature such as SMiRL. (Berseth et al., 2019)

## 2. Experiment Setup

This study assessed strategic evasion behaviors in mice through a task approximating predator-prey interactions within a hexagonal arena, namely "cellworld" (Lai et al., 2024). The physical setup included a custom robotic predator-like threat that pursued the mice, simulating predatory threat through aversive air blasts, while the mice attempted to reach a designated reward area without being "captured" (within 27.5 cm or 0.1 units of the robot). The arena was mapped with a hexagonal grid of 331 magnetic cells, each 11 cm apart, where obstacles were strategically placed to encourage adaptive evasion strategies(Mugan & MacIver, 2020).

To record precise positioning and behavior, a high-speed tracking system monitored both the robot and mice at 90 Hz,

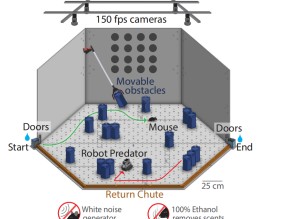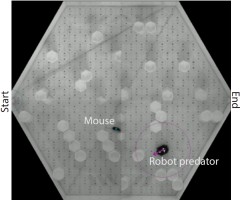

*Figure 1.* Real Mouse Setting

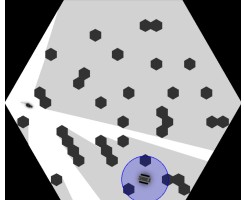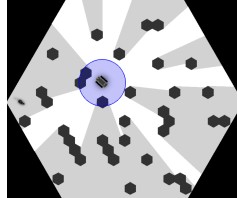

*Figure 2.* RL Setting: Mouse's view (left), predator's view (right).

facilitating real-time adjustments for the robot's trajectory through a combination of $A^*$ pathfinding and PID control. The robot initiated each episode from a starting position outside the mouse's field of view and actively navigated the arena by algorithm 1. The experimental arena featured both rewards and threats: water rewards were available at specific locations (Figure 1, doors), while mice had to simultaneously avoid being caught by the pursuing robot. The mice were water-restricted, which motivated them to explore the arena despite the predator's presence. The experiment involved eight lab mice (4 male and 4 female), each performing multiple 30-minute sessions daily. This data collection procedure follows previously established methodologies (Lai et al., 2024).

---

**Algorithm 1** Autonomous robot predator behavior

---

1: **while** experiment is running **do**
2:     Find spawn cell $S$
3:     Move robot to $S$
4:     **while** episode is running **do**
5:         **if** mouse is visible **then**
6:             Move robot to last seen mouse cell
7:         **else if** mouse is not visible **then**
8:             Find cells not visible to robot
9:             Randomly select a non-visible cell $N$
10:            Move robot to $N$
11:         **end if**
12:     **end while**
13: **end while**

---

For a direct comparison with the reinforcement learning (RL) model, we created a simulated environment, "cellworld gymnasium", building upon the Gymnasium API (Towers et al., 2024) and Pygame, which closely mirrors the physical experiment setup. This simulated environment replicates the hexagonal grid layout of the real arena and incorporates

the same reward structure: the virtual "mouse" receives a reward of +1 upon reaching the goal without capture and a penalty of -1 if it is "puffed" by the simulated predator-like robotic threat.[2] To further ensure fidelity, both the mouse and the predator in the RL environment move at speeds proportionally scaled to the average speeds observed in the physical trials. Additionally, the RL agent has partial observations, detecting the predator's location only when the predator is within its field of view. All distances within the arena are normalized such that 1 unit corresponds to the diameter of the hexagonal arena.

# 3. Reinforcement Learning (RL)

Reinforcement Learning (RL) provides a robust framework for sequential decision-making (Sutton et al., 1998), where an agent learns to maximize cumulative rewards through interactions with an environment. Formally, the RL problem is defined as a Markov Decision Process (MDP), characterized by the tuple $(\mathcal{S}, \mathcal{A}, \mathcal{P}, r, \gamma)$. Here, states $\mathcal{S}$ represents all possible states, actions $\mathcal{A}$ denotes available actions, transition dynamics $\mathcal{P}(s'|s, a)$ defines the transition probability from state $s$ to $s'$ given action $a$, $r(s, a)$ specifies the reward for taking action $a$ in state $s$, and discount factor $\gamma \in [0, 1)$ balances future versus immediate rewards.

The objective of RL is to find an optimal policy $\pi^*(a|s)$ that maps states to action distributions, maximizing the expected cumulative discounted reward:

$$J(\pi) = \mathbb{E}_\pi \left[ \sum_{t=0}^\infty \gamma^t r(s_t, a_t) \right]$$

## 3.1. Online RL: Learning through Interaction

Online RL involves the agent iteratively interacting with the environment to gather data and refine its policy. This paradigm is further divided into:

### 3.1.1. MODEL-FREE METHODS

These methods optimize the policy directly without constructing an explicit model of the environment's dynamics. Notable examples include:

**Deep Q-Network (DQN)**: It utilizes a neural network to approximate the action-value function $Q(s, a)$. The update rule is based on the Bellman equation (Hester et al., 2018):

$$Q(s, a) \leftarrow r(s, a) + \gamma \max_{a'} Q(s', a').$$

**Soft Actor-Critic (SAC)**: An actor-critic (Konda & Tsitsik-

---

[2]The equal magnitude (+1/-1) for rewards and penalties was chosen to reflect that both water reward and air blasts are relatively mild stimuli.

lis, 1999) algorithm that simultaneously optimizes a policy and a value function by maximizing a trade-off between expected reward and entropy ($\alpha$: entropy coefficient), promoting exploration (Haarnoja et al., 2018):

$$J(\pi) = \mathbb{E}_{(s,a)\sim\pi} \left[ r(s, a) - \alpha \log \pi(a|s) \right].$$

### 3.1.2. MODEL-BASED METHODS

Model-based methods learn a world model $\mathcal{P}(s'|s, a)$ to predict environment dynamics (Ha & Schmidhuber, 2018; Okada & Taniguchi, 2022). TDMPC-2 combines temporal difference (TD) learning with model predictive control (MPC) (Morari & Lee, 1999; Nagabandi et al., 2018), using the world model's trajectories to jointly optimize policies and value functions (Hansen et al., 2022; 2023). The algorithm operates by first using an encoder $z = h(s)$ to transform states into latent representations, then employing latent dynamics $z' = d(z, a)$ to predict the next latent state. It includes a reward predictor $r^* = R(z, a)$ for estimating immediate rewards and a value predictor $q^* = Q(z, a)$ for predicting future rewards in the latent space. A policy prior $\hat{a} = p(z)$ proposes actions to maximize the predicted value. This hybrid approach improves both sample efficiency and policy performance.

TDMPC-2 optimizes two objectives. The model objective $\mathcal{L}(\theta)$ minimizes the discrepancy between predictions and actual values:

$$\mathcal{L}(\theta) = \mathbb{E}_{(s,a,r,s')\sim\mathcal{B}} \left[ \sum_{t=0}^H \lambda^t \Big( \|z'_t - \text{sg}(h(s_t))\|_2^2 \right.$$
$$\left. + \text{CE}(r^*_t, r_t) + \text{CE}(q^*_t, q_t) \Big) \right]$$

where $\mathcal{B}$ denotes the replay buffer and CE is the cross-entropy loss. The policy objective $\mathcal{L}_p(\theta)$ maximizes returns while maintaining exploration:

$$\mathcal{L}_p(\theta) = \mathbb{E}_{(s,a)_{0:H}\sim\mathcal{B}} \left[ \sum_{t=0}^H \lambda^t \left( \alpha Q(z_t, p(z_t)) - \beta \mathcal{H}(p(\cdot|z_t)) \right) \right]$$

For action selection, TDMPC2 uses MPPI (Williams et al., 2017) to optimize over a horizon $H$:

$$\mu^*, \sigma^* = \arg\max_{(\mu,\sigma)} \mathbb{E}_{\mathcal{N}(\mu,\sigma^2)} [\gamma^H Q(z_{t+H}, a_{t+H})$$
$$+ \sum_{h=t}^{H-1} \gamma^h R(z_h, a_h)]$$

# 4. Comparison Between Mouse Behavior and RL Agent Behavior

By directly comparing mouse behavior with RL agents in the "cellworld" setting, we aim to characterize key behavioral

differences in their threat evasion strategies. Specifically, we analyze trajectory patterns to quantify wall-following tendencies, examine state density distributions to understand waiting behaviors near obstacles and starting points and compare action sequences before and after predator detection. Through these analyses, we seek to uncover the fundamental differences between mice and RL agents in risk management during predator-prey interactions.

We begin by training TDMPC-2 on the "cellworld gymnasium" environment and comparing the behaviors generated to those of the mice. On a superficial level, the performances of the mice and TDMPC-2 are similar, with the mice achieving an 86% success rate[3] across environments and the RL algorithms achieving about 80%. However, when we examine the exact nature of these successes, we see there is a dramatic divergence between the two agents.

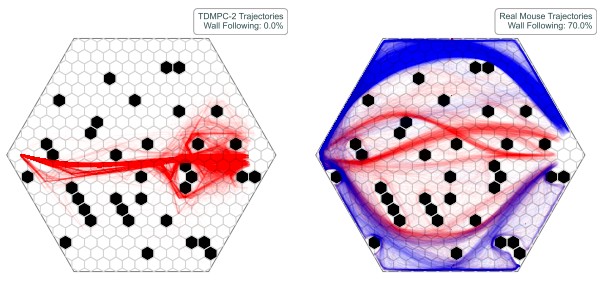

*Figure 3.* Trajectory plots: RL (left) vs Mouse (right). Blue indicates wall-following trajectories (thigmotaxis), while red indicates non-wall-following trajectories.

We also find that on average RL agents explore only 21.3% of the available space. This is in contrast to mice, which visit 77% of the entire arena. The visitation pattern overlap between RL agents and mice is only 20.9%.

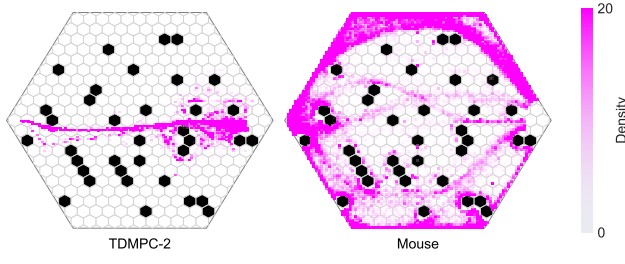

*Figure 4.* Density plot of RL (left) vs Mouse (right)

As shown in Figure 3, mice trajectories exhibit distinct patterns concentrated along walls (thigmotaxis, commonly observed in many animals and corresponding to measures of anxiety (Mugan & MacIver, 2020) and obstacles, with approximately 70% of trajectories following wall-following paths.[4] In contrast, RL agents' trajectories show direct,

---

[3]Proportion of runs in which the agent reached the goal without being captured.

[4]Defined as trajectories where over 70% of states are within

goal-oriented paths with minimal environmental interaction, with 0% of trajectories hugging walls. The state-visitation density of RL agents is heavily concentrated along these direct routes, showing little of the preemptive caution or environmental awareness displayed by the mice.

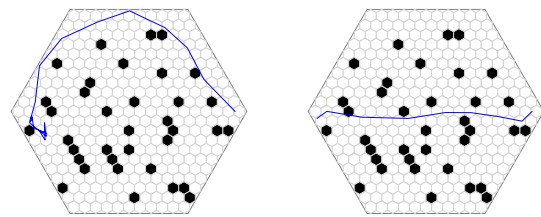

*Figure 5.* An example of distinct behaviors: Mouse (left) exhibits hesitation (dense segments near entrance), while RL agent (right) takes a direct path. Additional examples are in the Appendix.

This behavioral disparity is particularly evident in situations where the predator is not visible. As illustrated in Figure 5, mice exhibit distinct waiting behavior, shown by dense trajectory segments near the entrance (left), while the RL agent takes a direct path through the arena (right). This waiting behavior is further confirmed by the density plots in Figure 4. RL and mice agents start from the same position, but only the mice show high-density concentrations near the entrance, obstacles, and walls. While mice tend to wait and observe in such scenarios, RL agents maintain their direct approach, suggesting a fundamental difference in risk assessment and strategic planning between biological and artificial agents.

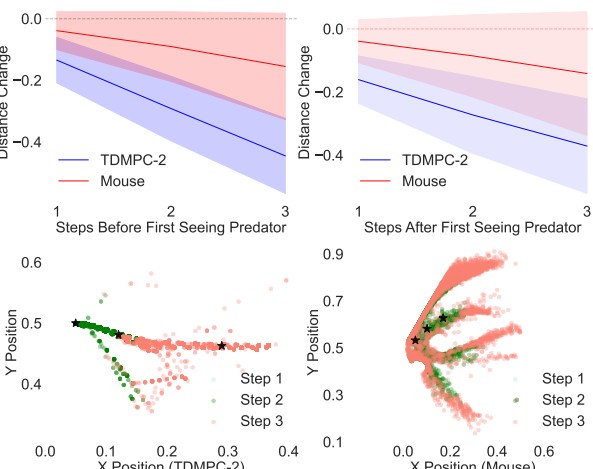

*Figure 6.* Top row: Mean changes in goal-agent distance before and after predator detection (left, right), with shaded areas showing standard deviation. Bottom row: spatial distribution of first three steps for each agent, with black stars showing mean positions. (1 unit = the diameter of the hexagonal arena)

Figure 6 highlights key behavioral differences between

---

0.1 units (10% of the diameter of the hexagonal arena) of the wall.

TDMPC-2 and mice. In reduction of distance to goal (top row), TDMPC-2 shows excessive movement both before (-0.2 to -0.4) and after (-0.2 to -0.35) predator detection, while mice maintain conservative patterns (-0.05 to -0.15), reflecting natural cautiousness. Spatially (bottom row), mice trajectories exhibit structured, radial exploration patterns, whereas TDMPC-2's movements appear scattered and disorganized.

Another key difference lies in learning efficiency after negative encounters. RL agents typically require multiple air blasts before adapting their behavior to avoid the predator, while mice exhibit immediate and persistent avoidance after a single encounter. Of course, this disparity is natural—mice have millions of years of evolutionary risk avoidance encoded in their genetics, while neural networks begin learning from scratch. However, this gap highlights an important challenge in artificial intelligence: how to develop RL systems that can more readily incorporate risk-avoidance behaviors without requiring extensive negative experiences.

## 5. Changing RL Behaviors to Match Biology

### 5.1. Trauma-Inspired Safety Memory in RL Agents

Can we teach RL agents to value their own 'lives' more and have a greater fear of the predator?

A common approach is Prioritized Experience Replay (PER) (Schaul et al., 2015). Our experiments demonstrate that PER provides some benefits, but does not fully address the challenges in our scenario. While PER accelerates learning, it has minimal impact on the agents' behavioral policies. This limitation motivated the development of our novel extensions.

We introduce the Trauma-Inspired Safety Buffer (TISB), which modifies standard experience replay to mimic how biological systems learn from trauma (Al Abed et al., 2020). In nature, rodents overweight negative experiences during risk assessment (LeDoux, 2013). Our approach implements this through two mechanisms:

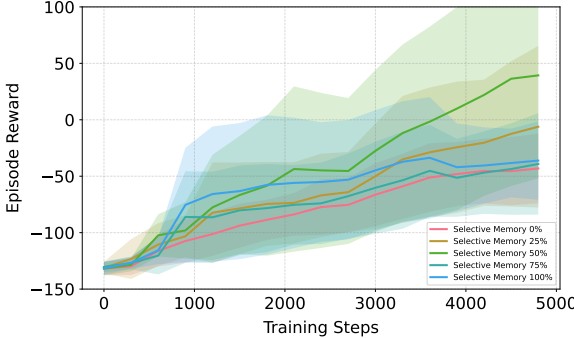

Figure 7. Experiments over 5000 steps show that 50% negative experiences yield optimal performance (5 random seeds).

Threat Experience Rebalancing: Inspired by how rodents show heightened sensitivity to dangerous situations, we investigate whether a higher proportion of negative experiences is necessary for effective learning. We employ various percentage settings to ensure that each training batch contains a specified proportion of negative transitions during sampling. Our experiments (Figure 7) shows that 50% negative experiences in the training batch yields optimal performance. This finding provides valuable insights into the importance of negative experiences during early learning phases, similar to how young animals develop risk assessment behaviors.

Threat Experience Amplification: The rewards of negative experiences are amplified by a factor of 200 during sampling[5], reflecting the heightened emotional intensity of trauma in biological systems, which leads to stronger behavioral changes.

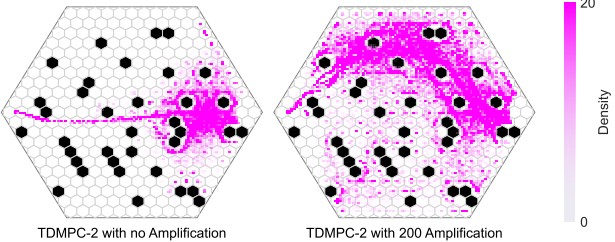

Figure 8. Standard TDMPC-2 with limited exploration patterns; Right: With a Threat Experience Amplification factor of 200, showing increased path diversity and enhanced wall-following behavior.

Our computational experimental results demonstrate that the TISB induces more naturalistic behaviors in RL agents. In our long-term experiments, TISB was implemented primarily through the Threat Experience Amplification mechanism. Figure 8 illustrates broader exploration and wall-following patterns similar to those observed in mice. However, the agents still lack the characteristic waiting behaviors seen in mice, as evidenced by the significantly lower density at the entrance compared to mice. This observation motivates our next approach.

### 5.2. Learning to Wait Through Variance-Penalized Temporal Difference (TD) Learning

Our analysis of mouse behavior reveals that pausing before potentially surprising situations is a key survival strategy. Mice tend to wait and gather information when they cannot see predators or when entering new areas. This observation suggests that effective survival behaviors might emerge

---

[5]We conducted experiments with different amplification factors: smaller values showed limited behavioral changes, while factors above 200 slowed down training without providing substantial benefits.

from an agent's desire to encode uncertainty in decision-making circuits (Rushworth & Behrens, 2008) and minimize surprise in its environment. One established framework for achieving such behavior is Surprise Minimization in Reinforcement Learning (SMiRL) (Berseth et al., 2019), which explicitly encourages agents to seek predictable states.

To embed SMiRL in TDMPC-2, we assign the intrinsic reward

$$r_{\text{SMiRL}}(s_t) = -\log p(s_t \mid s_{<t}),$$

which penalizes unlikely states under the agent's learned history. Estimating the full conditional density $p(s_t \mid s_{<t})$ is prohibitively costly and numerically brittle in our high-dimensional, rapidly changing environment, so we approximate the negative log-likelihood with the mean-squared prediction error in the model's latent space. Under the common fixed-variance Gaussian assumption, the MSE differs from the negative log-likelihood by only an additive constant (Girin et al., 2019), preserving the intuition that large prediction errors signal rare or "surprising" states. Because the latent representation learned by TDMPC-2 is especially sensitive to abrupt events—such as a predator suddenly entering view—the MSE provides a responsive and computationally lightweight proxy for surprise, yielding a stable and efficient realization of the SMiRL objective.

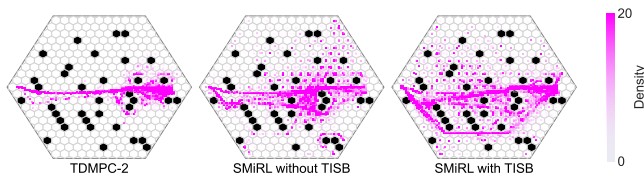

*Figure 9.* Density plots: TDMPC-2 (left), SMiRL without TISB (middle), and SMiRL with TISB (right). SMiRL may negate TISB's effect, as indicated by only minor changes in the main density regions.

Unfortunately, incorporating SMiRL into TDMPC-2 did not have the desired effect of making RL agents more conservative and cautious. As shown in Figure 9, while SMiRL shows some increased density in the central area compared to the baseline TDMPC-2, suggesting slightly more diverse path choices, the core trajectory pattern remains largely unchanged, as evidenced by the similar high-density regions (bright pink areas) across all conditions. Most importantly, SMiRL failed to demonstrate the crucial waiting behavior at the entrance, which was our primary goal. Furthermore, when combined with TISB, SMiRL appears to counteract TISB's path-finding benefits that were previously observed in TDMPC-2, resulting in only minor changes to the density distribution compared to the more substantial strategy shifts we were aiming for.

We attribute these limitations to SMiRL's fundamental mechanism of penalizing surprise across all state transitions. In our partially observable environment, where states naturally fall into distinct risk categories (e.g., safe vs. dangerous), this uniform penalization becomes problematic. By treating all transitions equally, SMiRL may inadvertently discourage beneficial state transitions, such as moving from dangerous to safe states, and interfere with TISB's path-finding capabilities.

To address these issues, we propose a more selective extension approach based on our observation that Q-value variance effectively signals states requiring caution. Our empirical analysis shows that Q-value variance across actions spikes sharply when the predator appears or approaches, naturally indicating states where cautious behavior is crucial. This insight leads us to develop a variance-penalized TD target that selectively adjusts penalties based on the agent's uncertainty in different states. We formalize this approach as:

$$TD_{\text{target}} = r_t + \gamma \left( Q(s_{t+1}, \pi(s_{t+1})) - \alpha \text{Var}_{a \in A}(Q(s_{t+1}, a)) \right)$$

Here, $\alpha$ is a coefficient that controls the strength of the variance penalty. During training, we compute this variance by uniformly sampling a grid of actions from the action space, calculating Q-values for each action, and then computing the variance of these Q-values. This process results in a penalty term that increases in states where different actions lead to highly variable outcomes.

Our method diverges from SMiRL in two critical ways: instead of modeling state transition probabilities, we directly measure uncertainty through Q-value variance across actions, aligning penalties with the agent's decision-making confidence. Rather than statically penalizing all state transitions via rewards, we dynamically adjust penalties within the TD target computation, enabling risk-averse behavior to emerge contextually, particularly in high-uncertainty states such as predator proximity.

In summary, both approaches can be viewed as implementing different forms of uncertainty minimization:

$$\text{SMiRL:} \quad \max_{\pi} \mathbb{E}\left[ \sum_t \gamma^t \left( r_t - \alpha_1 \log p(s_t \mid s_{<t}) \right) \right]$$

$$\text{Our approach:} \quad \max_{\pi} \mathbb{E}\left[ \sum_t \gamma^t \left( r_t - \alpha_2 \text{Var}_a(Q(s_t, a)) \right) \right]$$

Consequently, our approach encourages the agent to avoid states with high decision uncertainty, effectively balancing exploration and risk aversion. Intuitively, it incentivizes behaviors such as cautious planning and avoiding unnecessary risks.

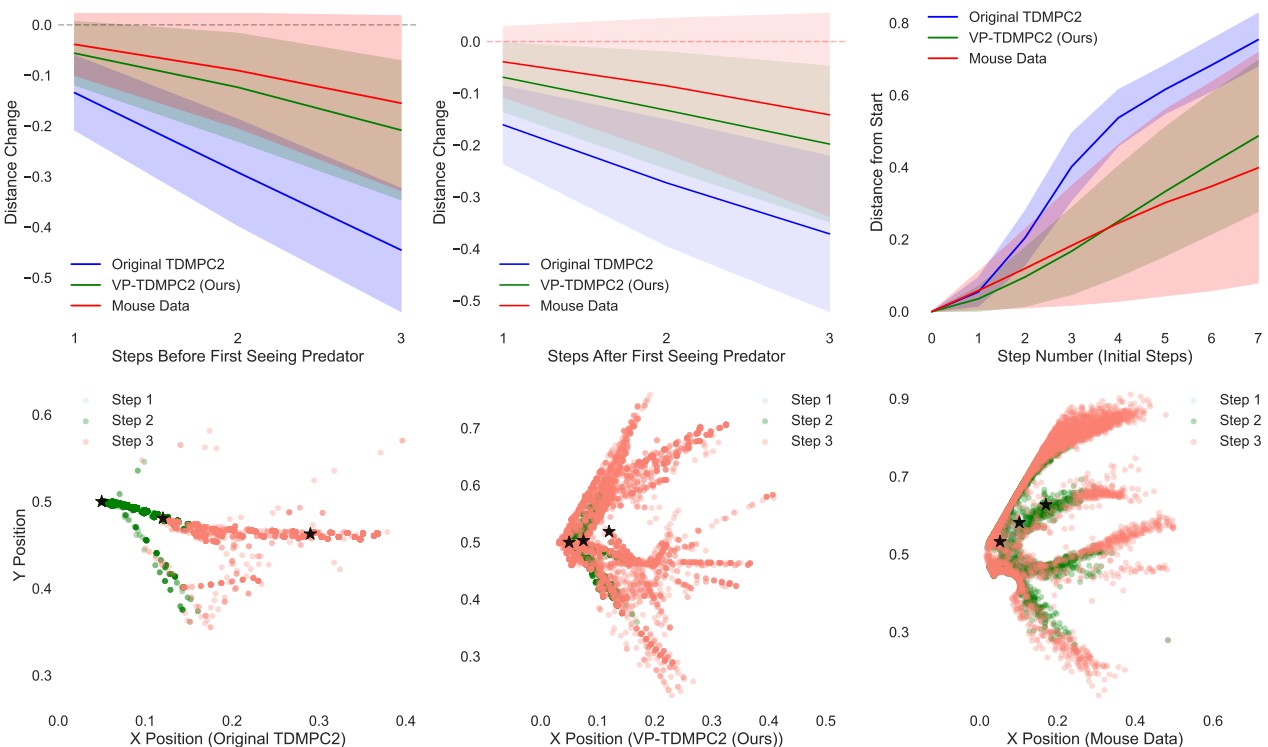

*Figure 10.* Top row: Mean changes in goal-agent distance before and after predator detection (left, middle), and initial mean movement distances (right). Shaded areas show standard deviation. Bottom row: Spatial distribution of first three steps for each agent, with black stars indicating mean positions. The black stars show that mouse data and VP-TDMPC2 maintain more consistent, cautious initial positions, while baseline TDMPC2's mean positions spread out more rapidly, indicating less conservative early behavior. (1 unit = the diameter of the hexagonal arena)

Figure 11 compares the three agents under investigation. Numerical calculations of the visitation pattern overlap show that the TDMPC-2 agent has a 20.9% overlap with mice, while VP-TDMPC-2 achieves 86.1%. This metric reflects similarity in exploratory coverage (whether both agents have been to the same states), rather than a one-to-one match in how frequently or in what manner they visit those states.

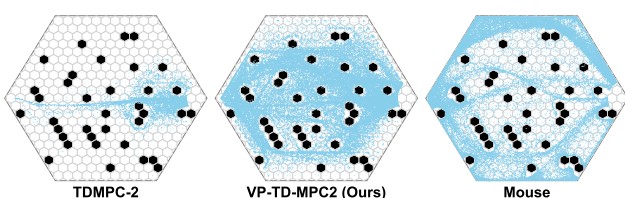

*Figure 11.* Final states visitation pattern comparison of original TDMPC-2 (left), to our VP-TDMPC2 (middle: trained with TISB and Variance-Penalized TD learning) and Mouse (right).

Notably, VP-TDMPC-2 exhibits geometric patterns and balanced coverage resembling mice's systematic exploration, though it lacks the extreme thigmotaxis (wall-following with minimal distance throughout the entire trial) observed

in real mice. For a closer examination of the differences between mice behavior and RL, we analyzed the initial part of the trajectory from the start gate.

Our analysis (Figure 10) reveals behavioral parallels with mice exploration patterns. Before encountering the predator, VP-TDMPC-2 demonstrates natural hesitation patterns, with distance changes closely matching mice behavior (-0.1 to -0.2), while the baseline TDMPC-2 shows excessive movement (-0.2 to -0.5). The spatial distribution plots (bottom row) further illustrate that our method captures key characteristics of mice behavior: initial hesitation near entry points[6], some thigmotactic movement along boundaries, and structured radial exploration patterns.

To further quantify these behaviors, we conduct additional analyses focusing on two metrics: waiting behavior[7] and episode length. Across 1,000 trajectories, standard TD-MPC showed 0% incidence of waiting, while real mice

---

[6]Trajectory comparisons (Figure 15) in the appendix provide additional evidence of the behavioral similarity between VP-TDMPC-2 and real mice.

[7]Defined as movement with distance changes under 0.1 units within the first six steps.

exhibited about 27.7% and our VP-TDMPC-2 achieved about 32.4%.

Episode length also differed substantially (Figure 13 in Appendix). Standard RL agents completed the task in an average of 7.09 steps, while VP-TDMPC-2 averaged 13.89 steps, closely aligning with the 14.04-step average observed in mice. Longer episodes may reflect more cautious or deliberate planning under risk.

Together, these quantitative findings suggest that our proposed mechanisms enhance the agent's ability to model biologically plausible risk-sensitive behaviors.

### 5.3. Challenges in Closing the Mice and Machines Gap

As shown in Figure 11, the final converged policies still exhibit qualitative differences. Mice display stronger wall-following behavior, whereas our improved RL agents tend to make greater use of the central regions.

Our hypothesis is that this behavioral difference arises from the structure of the RL state space. After thoroughly exploring the outer wall during training, the agent no longer needs to maintain constant wall contact for safety. Upon reaching the top area of the arena, it can confidently conclude that the task is solved. Additionally, the original TDMPC2 agent relied almost entirely on the central regions. Although our improved RL agent has learned to exhibit some cautious, mouse-like behavior, it has also learned that the central area offers more efficient paths to the goal when safety is ensured. In contrast, mice continue to follow the wall regardless of task mastery, likely due to rodents' innate behavioral priors (Champagne et al., 2010).

We observe that performance can be sensitive to the penalty-weight hyperparameter: setting it higher may encourage overly cautious policies, whereas lower settings can sometimes produce risk-prone behavior. Moreover, our predator–prey environment is deliberately highly stochastic—even with fixed seeds—to better mimic real predator randomness, which can occasionally lead to less stable training compared to standard RL benchmarks. These limitations highlight the broader challenge of aligning artificial behaviors with natural ones, both in terms of behavioral fidelity and parameter robustness.

## 6. What about LLM agents?

Given the remarkable capabilities of transformer models in sequence modeling and reasoning (Brown et al., 2020), we are particularly interested in examining how a large language model (LLM) agent behaves in a controlled environment. Recent work has shown that LLMs like GPT-4 can exhibit behavior consistent with human-like theory of mind in controlled tasks (Strachan et al., 2024), raising impor-

tant questions about the extent and nature of their internal representations. Does the behavior of an LLM resemble that of mice, or is it closer to that of an RL agent? Such an experiment may provide insights into the world model of LLMs and their biological alignment. In this study, we use ChatGPT-4 to perform the experiments.

Parallel to the RL setting, the LLM agent receives state information in two forms: a visual representation of the current environment and a textual description of the task. The LLM agent operates under the same partial observability constraint, where the predator's location is only revealed when it falls within the agent's field of view. Our LLM agent makes decisions at each timestep, identical to TDMPC2, ensuring a fair comparison. To maintain experimental integrity and assess the LLM's inherent capabilities, the prompts are designed to be neutral without any suggestive cues about optimal strategies.

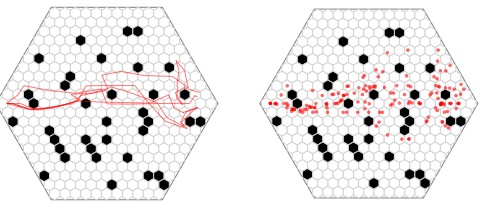

*Figure 12.* LLM agent trajectories (left) and scatter plots (right). Experiments were run with fewer trials due to API cost constraints, but this was sufficient to illustrate the observed pattern.

Interestingly, like RL agents, the LLM agent often exhibits reckless behavior (Figure 12). At the beginning of each trial, the agent enters the environment without apparent strategic considerations, despite expressing intentions such as: "Moving cautiously towards the goal while avoiding obstacles and staying alert to the unseen predator's movements."

When encountering the predator, the agent does take evasive actions, with internal reasoning resembling: "Moving horizontally to avoid obstacles and maintain distance from the predator while progressing toward the goal." However, the observed behavior significantly differs from that of mice. While the LLM agent demonstrates reactive adjustments to visible threats, its overall approach lacks the nuanced, adaptive strategies often exhibited by biological agents like mice. We believe this suggests the internal world model of LLMs might lack biological alignment along certain axes, particularly regarding anxiety and fear.

## 7. Related Work

Reinforcement learning has emerged as a fundamental framework for understanding decision-making processes in neuroscience and bio-behavioral research (Botvinick et al., 2020; Subramanian et al., 2022; Fan et al., 2023). Researchers (Schultz et al., 1997; Niv, 2009) discovered that

dopamine neuron activity patterns in the midbrain closely mirror the reward prediction error signals central to temporal difference learning. Neuroimaging studies (Lockwood & Klein-Flügge, 2021) have revealed neural correlates of Q-values in the ventral striatum and orbitofrontal cortex, establishing these regions as key components in value-based decision-making.

The discovery of mirror neurons, which exhibit similar activation patterns during both action execution and observation (Buccino et al., 2004), has provided a biological foundation for imitation learning, influencing both robotic systems (Zahra et al., 2022) and third-person imitation learning algorithms in RL (Stadie et al., 2017). Building on biological inspiration, researchers have also integrated curiosity-driven learning mechanisms (Stadie et al., 2015; Oudeyer & Smith, 2016) into RL algorithms to enhance exploration and learning efficiency. Recent advances in distributional reinforcement learning suggest that the brain encodes not merely the mean expected reward, but rather the complete distribution of possible rewards (Muller et al., 2024).

Comparative studies have increasingly focused on aligning biological and artificial behavior. For instance, Vaxenburg et al. (2024) and Singh et al. (2023) analyze navigation tasks across species and models. Our use of model-based RL draws on findings by Daw et al. (2011), who demonstrated that animals employ internal models in decision-making, with striatal signals resembling TD prediction errors. Furthermore, Mattar & Daw (2018) showed that prioritized memory access can explain biological planning, which directly motivates our TISB framework. In parallel, Blanchard et al. (2011) identified neural mechanisms for risk assessment in rodents, aligning with our variance-penalized objective. Together, these studies suggest that while standard RL falls short of replicating mouse-like behavior, biologically grounded modifications offer a promising path toward closing this gap.

## 8. Discussion

In this paper, we present a detailed parallel comparison between machines and mice in an environment characterized by risks, goals, and obstacles. The parallel comparison allows us to explore behavioral differences in depth: mice show distinct path-planning strategies, often sacrificing path efficiency for safety, while RL agents gradually learn to follow distance-optimal routes. Most importantly, we observe waiting and peeking behaviors in mice at entrances, behind obstacles, and other locations where the predator is not visible, while regularly trained RL agents never exhibit such behaviors.

These behavioral discrepancies may arise from a fundamental architectural gap. Unlike biological agents that evolve in the context of decisions where the outcome is irreversible (succumbing to a predator, for example), standard RL systems lack an explicit notion of mortality (Ororbia & Friston, 2024). Without permanent failure states, RL agents tend to over-explore and underweight rare but fatal risks, prioritizing immediate goals over the survival-oriented caution observed in biological threat responses. Irreversibility in outcomes has motivated using imagination to inform policy learning in planning algorithms (Racanière et al., 2017), and is a possible basis for the evolution of imagination and planning in the brain (MacIver & Finlay, 2022).

To bring RL agents closer to mice, we introduce two mechanistic innovations: the Trauma-Inspired Safety Buffer and variance-penalized TD learning. The TISB's dual mechanisms provide key insights: Threat Experience Rebalancing reveals the optimal proportion of negative experiences during early learning, while Threat Experience Amplification enhances wall-following behaviors similar to mice's defensive strategies. Meanwhile, the variance penalty objective induces information-gathering behaviors reminiscent of biological risk assessment. These mechanisms not only make agent behaviors more mice-like but also advance the interpretability of RL systems by grounding their decision-making processes in biologically plausible behaviors.

We do admit that our work indeed focuses on a specific predator-prey domain, and the proposed mechanisms were intentionally designed within this context. However, predator-prey dynamics represent fundamental survival interactions observed across both natural and artificial systems, as highlighted by Marrow et al. (1996) and Tsutsui et al. (2024). These studies emphasize such scenarios as key paradigms for investigating adaptive behavior. Importantly, we view this work as an exploratory step toward aligning artificial agent behavior with biologically inspired patterns, rather than as a definitive benchmark for performance or generality.

Several promising directions emerge from our findings. While our methods improved wall-following behavior, certain patterns—like extreme wall-kissing trajectories—remain challenging to replicate. The dynamic predator presence complicated our attempts with intrinsic rewards, suggesting the need for novel architectures to capture these biological strategies. An observed "baiting" behavior is of special interest—where mice intentionally attract predator attention before retreating (Lai et al., 2024). While TDMPC-2 incorporates planning horizons, replicating such deceptive strategies requires deeper integration of opponent modeling, pointing to opportunities to develop better prediction mechanisms.

## Acknowledgments

We thank the anonymous reviewers for their helpful feedback in clarifying our conclusions about generalization, shaping our motivation, guiding our approach to quantifying risk awareness, and improving the related work section.

## Impact Statement

This paper presents work whose goal is to advance the field of Machine Learning. While there are many potential societal consequences of our work, we want to highlight the following: All experiments involving eight laboratory mice (4 male/4 female) employed 30-minute behavioral sessions designed to minimize stress, strictly following NIH humane treatment standards. Experimental procedures were in accordance with NIH guidelines and approved by the corresponding institutions' Animal Care and Use Committee.

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

## A. Environment Details

In our environment setting, we employ both discrete and continuous action spaces, tailored for position control. The continuous action space is utilized for algorithms such as SAC and TDMPC-2, while the discrete action space is designed for algorithms like DQN or other discrete action-based methods.

In the continuous action space, we have a three-dimensional action space. The first two dimensions represent the target coordinates (x, y) within the arena, indicating the position the agent should move to. The third dimension serves as a waiting or movement indicator: if this value is larger than 0.5, the agent should not move. The agent learns the value of these actions through interaction with the environment.

In the discrete action space, the environment is modeled using a hexagonal grid, where the arena is a large regular hexagon composed of smaller hexagonal cells. Each cell is assigned a unique identifier, effectively discretizing the action space. In this setting, each step corresponds to 0.25 seconds in the real-world mouse experiment.

The state space is represented as a ten-dimensional vector, which includes the following information:

- The simulated mouse's own position, including its $x$ and $y$ coordinates and the direction it is facing.
- The simulated robot's position, but only when the robot is visible to the mouse.
- The distance to the goal.
- A binary indicator of whether the mouse has been "puffed" by the robot.

The reward structure is sparse. All distances within the arena are normalized such that 1 unit corresponds to the diameter of the arena. This normalization ensures consistency and scalability across different experimental setups.

Upon each environment reset, the predator is initialized at a random location outside the prey's field of view. This initialization remains stochastic even when a fixed random seed is used, introducing variability in the predator's starting state across episodes. After initialization, the predator then begins hunting the prey following the procedure outlined in Algorithm 1.

## B. Experimental Details

All experiments were conducted with a maximum episode length of 300 steps and a total of 100,000 training steps. We use a timestep of 0.25s. This aligns with findings showing that rodents make approximately 2-5 decisions per second during navigation (Resulaj et al., 2009). Through environment setup, we determined that the 0.25s provides an optimal balance between biology realism and learning performance.

### B.1. Model-Free Methods

Experiments on SAC and DQN were implemented based on stable-baseline3 (Raffin et al., 2021). For baseline comparisons, DQN was implemented with an epsilon-greedy exploration strategy, using a learning rate of 1e-4 and batch size of 256. SAC was initialized with 1,000 random seed steps for exploration, trained with a learning rate of 3e-4 and batch size of 256. Both algorithms employed two-layer networks with 256 units per layer and parameters updated every step.

### B.2. Main Model-Based Method

For our main methods, TDMPC-2 and VP-TDMPC-2, we implemented both experiments based on the original code (Hansen et al., 2023). Both methods utilized 1,000 random seed steps for initial exploration and were trained with a learning rate of 1e-4 and batch size of 512. After training, we collected expert trajectories from TDMPC-2.

For VP-TDMPC-2, we modified the buffer and TD-target of the TDMPC-2 agents in their original code. To calculate Q-variance, we first sampled 400 actions uniformly from the action space. We used an ensemble of 5 different Q-networks, each predicting Q-values for these actions. The variance was then computed across all Q-values produced by these networks.

We used the horizon length of 3 is in our experiments. Our task mirrors real mouse behavior: excluding waiting periods, mice need 2-3 seconds to reach the goal. Given our simulation timestep (0.25 seconds per action), successful task completion in RL requires 8-12 steps. The gamma factor is 0.995.

The penalty coefficients in the range of 0.1–0.2 achieved the optimal balance between task completion and cautious behavior. For comparison with mice, we trained two separate agents with coefficients of 0.1 and 0.2, respectively, and collected expert

data from both. To prevent training instability and variance explosion, we implemented variance value clipping at 1,000 during VP-TDMPC-2's TD target updating.

As we have stated in the limitation section, while VP-TDMPC-2 demonstrated improved behavioral alignment with biological agents, we note that its training stability was limited. The model exhibited high variance, and successful cautious behavior only emerged under specific hyperparameter settings. Although our findings are representative, we emphasize that VP-TDMPC-2 remains a proof-of-concept method rather than a robust, general-purpose algorithm.

### B.3. Why not SAC and DQN

We experimented with DQN and SAC alongside TDMPC2 but focused on TDMPC2 because: (i) Model-based approaches together with TD learning better align with biological decision-making ([Daw et al., 2011](#)), (ii) DQN and SAC showed less stable convergence and inconsistent trajectories, and TDMPC2 produced more stable behaviors.

Notably, with sufficient training, all mentioned RL methods ultimately exhibited similar "reckless" behaviors (lacking waiting periods and showing minimal exploration), confirming that our observations about behavioral gaps are not specific to one algorithm.

### B.4. How we evaluate our agent

For the density plots presented in all sections, we select the best-performing model based on validation metrics to ensure that the comparison highlights the real exploration capabilities of each method. If we averaged multiple runs, one with density mainly at the top and another at the bottom, the result could wrongly suggest strong exploration, even if both actually explored poorly.

To ensure a fair comparison, both the baseline and improved models were trained following identical procedures, and the best-performing checkpoint (as measured by validation performance) was selected for each. The collected datasets are of similar scale to highlight the main patterns.

## C. Episode Length Distribution Plot

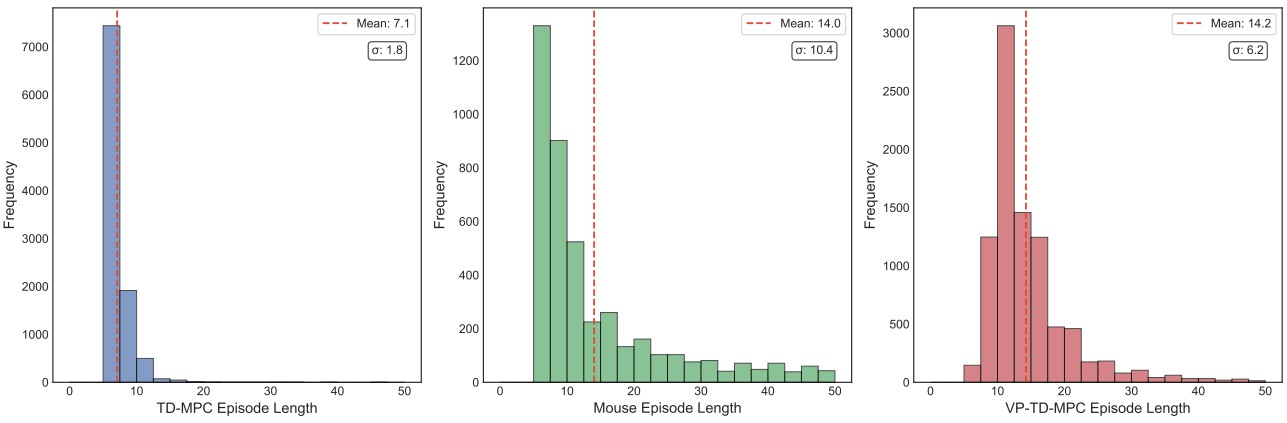

*Figure 13.* Comparison of episode length distributions across models and mice (filtered for episodes with length between 5 and 50 to exclude trivial failures and outlier trajectories unlikely to reflect meaningful exploratory behavior). TD-MPC2: Mean = 7.09, SD = 1.77, Range = [5, 46]; Mouse: Mean = 14.04, SD = 10.40, Range = [5, 50]; VP-TD-MPC2: Mean = 14.25, SD = 6.24, Range = [6, 50]; VP-TD-MPC2 reproduces mouse-like exploratory durations more faithfully than standard TD-MPC2, with a longer mean episode length and greater behavioral variability.

## D. More Example trajectories

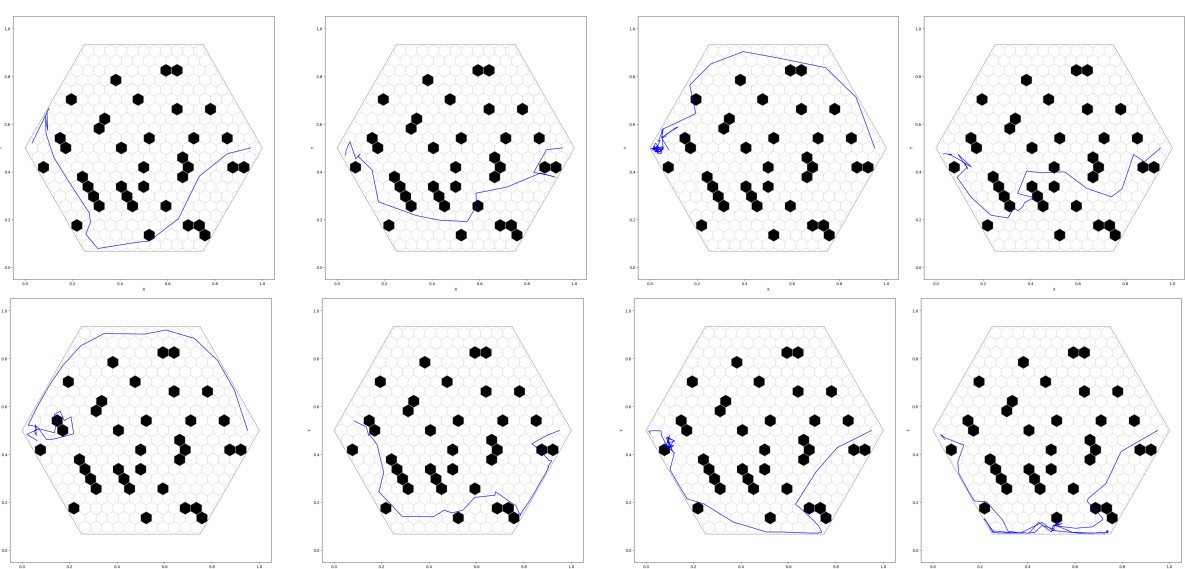

*Figure 14.* More mouse trajectories, demonstrating consistent waiting behavior at the entrance and strong thigmotactic patterns (wall-following behavior). We added slight noise to highlight the waiting path.

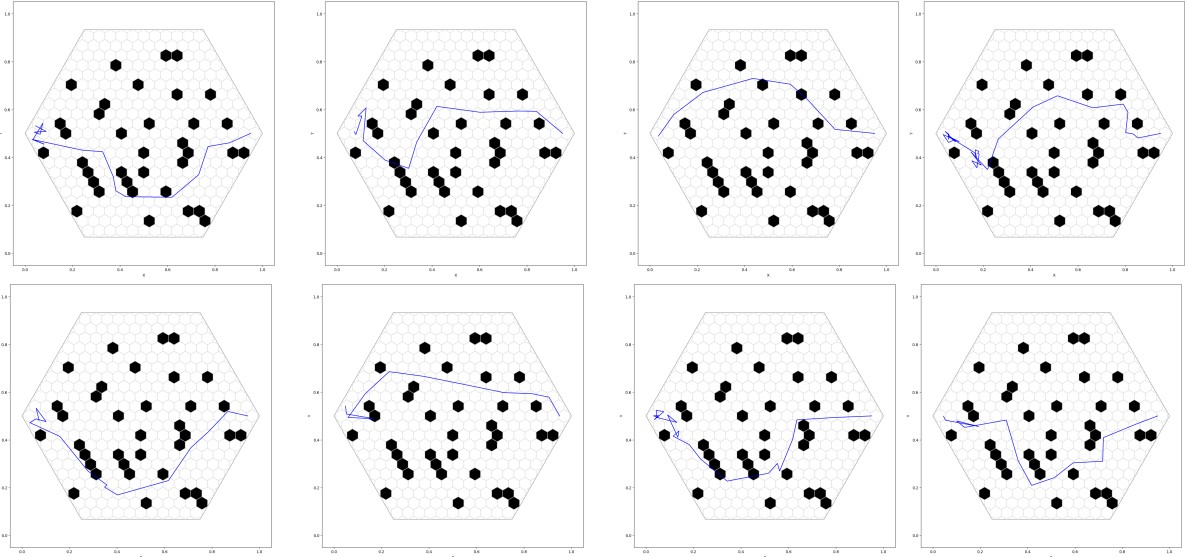

*Figure 15.* More VP-TDMPC-2 trajectories, exhibiting similar waiting patterns at the starting point and emerging thigmotactic tendencies, closely resembling mice behavior. We added slight noise to highlight the waiting path.

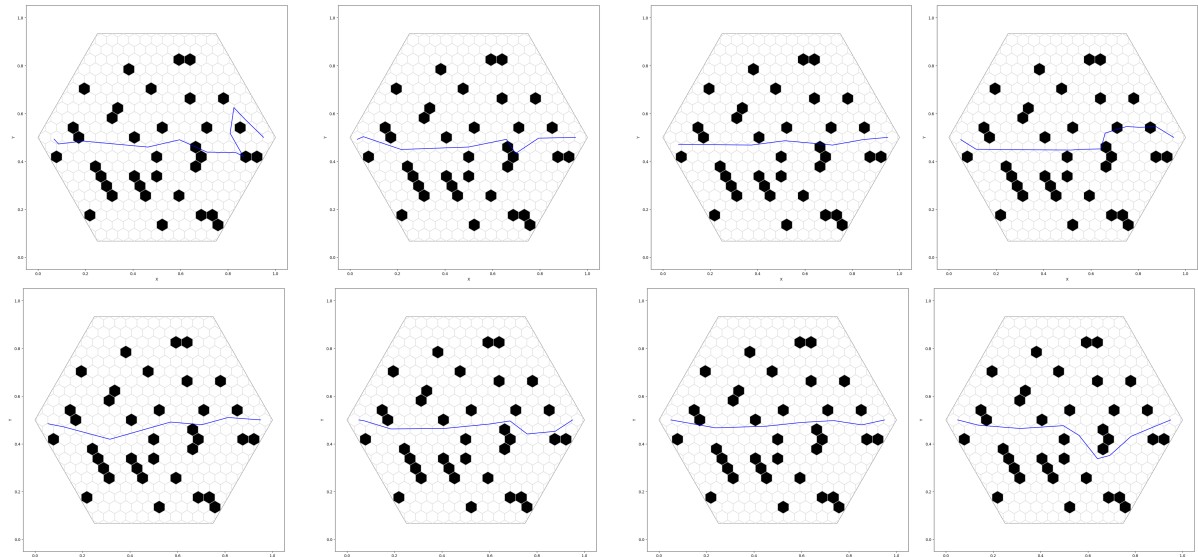

*Figure 16.* More TDMPC-2 trajectories, showing direct, goal-oriented paths with predator avoidance occurring only in close proximity to the goal, lacking the strategic caution observed in mice. We added slight noise to highlight the waiting path, otherwise it would be visually obscured due to overlapping trajectory lines.

## E. Fitting Offline RL On Mouse Data and Motivation for KL Divergence

To further validate the behavioral differences between mice and RL agents, we employ offline reinforcement learning to directly learn from mouse trajectory data. By training offline RL algorithms on mouse behavioral datasets, we can create "mouse-like" policies that mimic biological decision-making patterns.

Comparing these learned policies with standard online RL agents provides additional evidence for the fundamental differences in risk assessment and exploration strategies between biological and artificial agents.

Offline RL aims to learn optimal policies solely from pre-collected datasets without further environment interactions. This approach is essential in scenarios where live data acquisition is prohibitively expensive or risky. A primary challenge in offline RL is addressing the **distributional shift** between the data distribution $d_{\text{data}}(s, a)$ and the policy-induced distribution $d_\pi(s, a)$.

### E.1. Behavior Cloning (BC)

Behavior Cloning (Torabi et al., 2018) is a supervised learning approach that learns a policy $\pi(a|s)$ directly from the dataset by mimicking the actions taken by the behavior policy. The objective is to minimize the discrepancy between the learned policy and the behavior policy:

$$\min_\pi \mathbb{E}_{(s,a)\sim d_{\text{data}}} \left[ \ell \left( a, \pi(s) \right) \right],$$

where $\ell$ is a loss function measuring the difference between the actions.

**Limitations of BC**: While BC can effectively mimic certain aspects of mouse behavior, it struggles with capturing the underlying decision-making strategies, especially in states that are underrepresented or absent in the dataset. This limitation results in poor generalization and inability to solve the task.

## E.2. Offline RL Methods

To overcome the shortcomings of BC, more sophisticated offline RL methods modify standard RL objectives to account for the distributional shift:

1. **Conservative Q-Learning (CQL)**: CQL penalizes overestimation of Q-values for unseen actions (Kumar et al., 2020), ensuring the learned policy remains close to the data distribution. The objective is:

$$\min_Q \max_\mu \; \alpha\Big(\mathbb{E}_{s\sim d_{\text{data}}, a\sim \pi(a|s)}\big[Q(s,a)\big] \; - \; \mathbb{E}_{(s,a)\sim d_{\text{data}}}\big[Q(s,a)\big]\Big) + \mathcal{L}_{\text{Bellman}}(Q) + \mathcal{R}(\mu),$$

where $\mu$ is an auxiliary action-distribution and $\mathcal{R}(\mu)$ its regularizer.

2. **Conservative offline model-based policy optimization (COMBO)**: COMBO (Yu et al., 2021) extends CQL by incorporating a learned model of the environment. It utilizes model-based planning in conjunction with conservative value estimation to improve policy learning:

$$\min_Q \alpha \left(\mathbb{E}_{s\sim \hat{d}, a\sim \pi(a|s)}\left[Q(s,a)\right] - \mathbb{E}_{(s,a)\sim d_{\text{data}}}\left[Q(s,a)\right]\right) + \mathcal{L}_{\text{Bellman}}(Q),$$

where $\hat{d}$ represents the state distribution induced by the learned model.

3. **Implicit Q-Learning (IQL)**: IQL (Kostrikov et al., 2021) focuses on recovering optimal actions implicitly by decoupling the value function from policy improvement. It uses expectile regression to learn the value function:

$$\mathcal{L}_V(\psi) = \mathbb{E}_{(s,a)\sim d_{\text{data}}}\left[L_2^\tau\left(Q_\theta(s,a) - V_\psi(s)\right)\right],$$

$$\mathcal{L}_Q(\theta) = \mathbb{E}_{(s,a,r,s')\sim d_{\text{data}}}\left[\left(Q_\theta(s,a) - r - \gamma V_\psi(s')\right)^2\right],$$

where $L_2^\tau$ is the asymmetric squared loss with expectile $\tau$.

## Empirical Evaluation

In our predator-prey task, we evaluate the performance of different offline reinforcement learning methods in reproducing mouse behaviors. While both CQL and COMBO successfully reproduce the basic mouse trajectories, they fail to capture the nuanced waiting behavior observed in real mice. This limitation appears to stem from their overemphasis on reward optimization, which leads to overly aggressive strategies that deviate from the natural behavioral patterns.

In contrast, IQL demonstrates superior performance by accurately capturing both the waiting actions and the precise trajectory patterns exhibited by mice. This success highlights IQL's enhanced adaptability to complex behavioral dynamics and its ability to balance between reward optimization and behavioral fidelity.

Despite these promising results, the limited availability of mouse trajectory data in our dataset constrains our ability to learn a fully accurate mouse policy. This data scarcity represents a fundamental challenge that must be addressed in future work. We anticipate that developing more sophisticated behavior-cloning techniques will be essential for faithfully reproducing the subtle decision-making patterns observed in real mice.

To provide a quantitative assessment of the behavioral differences, we measure the discrepancy between the online RL policy and the mouse-derived policy using Kullback-Leibler (KL) divergence (Hershey & Olsen, 2007). For each state $s$ in a representative subset of the state space, the KL divergence between the two policies is defined as:

$$KL(\pi_{\text{IQL}}(\cdot|s)\|\pi_{\text{RL}}(\cdot|s)) = \sum_a \pi_{\text{IQL}}(a|s) \log \frac{\pi_{\text{IQL}}(a|s)}{\pi_{\text{RL}}(a|s)}$$

where $\pi_{\text{IQL}}(a|s)$ denotes the probability of selecting action $a$ given state $s$ under the IQL policy (representing mouse behavior), and $\pi_{\text{RL}}(a|s)$ represents the corresponding probability under the online RL policy. By averaging the KL divergence across all sampled states, we obtain an overall measure of policy divergence.

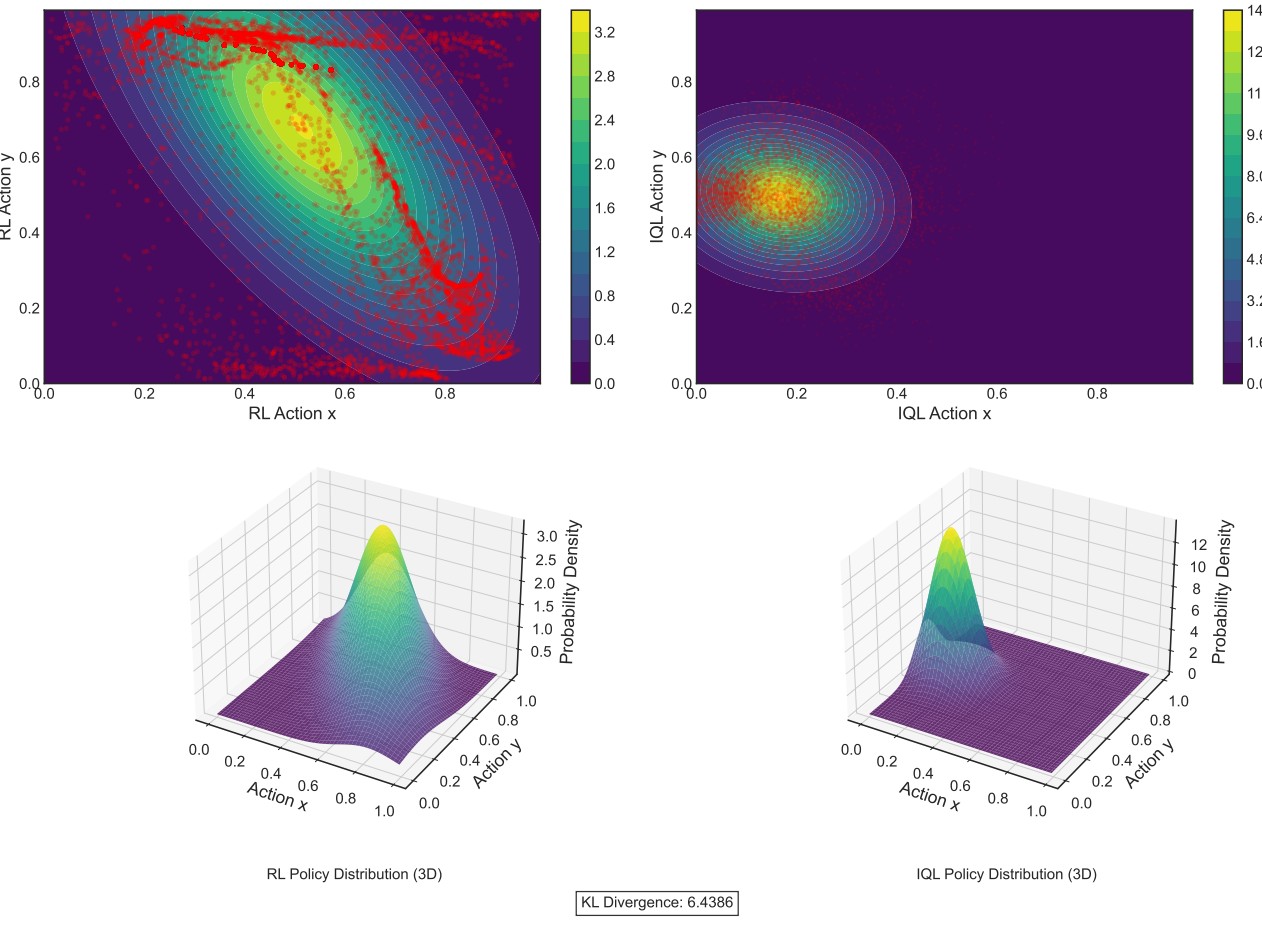

*Figure 17.* KL Divergence between online RL and IQL policies (IQL representing mouse behavior)

Our quantitative analysis reveals a substantial divergence between the online RL policy and the mouse-derived policy, with a KL divergence value of 6.4386 (Figure 17). This significant divergence underscores the considerable gap between current RL approaches and natural mouse behavior, highlighting the complexity of accurately modeling biological decision-making processes. These findings emphasize the need for more sophisticated approaches that can better capture the nuanced behavioral strategies employed by mice in predator-prey scenarios.

# F. ChatGPT-4o Experiment

*Table 1.* LLM Agent Prompt

You are controlling a prey agent in a predator−prey environment.

Your goal is to reach the green goal point (1.0, 0.5) while avoiding the predator and obstacles.

Environment Details:
− You (red dot) must reach the goal (green dot) while avoiding
the predator (blue dot)
− Black areas are obstacles/walls that you cannot pass through
− The predator (blue dot) is constantly moving and trying to catch you,
and there is a
larger blue circle indicating the puffed area around the predator
− If you can't see the predator, it means it's hidden behind obstacles
− The environment has a grid to help you locate positions
(x and y coordinates from 0 to 1)
− Each move must have an L2 norm less than 0.2

Your response must be a JSON object with exactly this format:
{
  "move": [
    {"x": <float>, "y": <float>}
  ],
  "thoughts": "<single line explaining your strategy>"
}

Rules for moves:
1. Provide exactly 1 move
2. Each move should be a small step
(L2 norm of distance between your next position and
current position < 0.2)

Example response:
{
  "move": [
    {"x": 0.20, "y": 0.45},
  ],
  "thoughts": "Moving toward the goal while avoiding obstacles and keeping
  distance from potential predator locations."
}

