# OpenReview forum: "Of Mice and Machines: A Comparison of Learning Between Real World Mice and RL Agents"
_ICML.cc/2025/Conference — ICML 2025 poster_

### Official Review · Reviewer_YZj9 · 2025-02-24

**Overall Recommendation:** 3

**Summary:**

This paper conducted and analyzed an experiment of "predator and prey" by mice and a robotic agent to learn the behavior patterns of biological self-preservation instinct, and designed and evaluated the simulation of reinforcement learning agents and the same "predator" robotic agent. The paper implemented negative memory amplification and variance-penalized temporal difference learning to introduce the risk-avoidance to reinforcement learning and decision making. The simulation results showed the two mechanisms made the reinforcement learning agents share more similar visitation pattern with the mice such as radial exploration and wall-following.

**Claims And Evidence:**

Yes

**Essential References Not Discussed:**

None to my knowledge

**Experimental Designs Or Analyses:**

Please refer to "Method and Evaluation Criteria" section

**Methods And Evaluation Criteria:**

1. Line 110 - 113, what is the meaning of "receives a reward of +1 upon reaching the goal", moving to the next cells safely? What is the meaning of "puffed by the simulated predator-like robotic threat"? It is fair to set 1 to both reward and penalty in the simulation to explain the difference between mice and RL agents, but I am wondering if increasing the penalty would make RL agents more cautious?

2. Line 172 - 179, what is the definition of success rate? Is it the same metric as "survival rate" first appeared in 2. Experiment Setup? What is the motivation of making RL agents move more like mice if the success rate is almost the same while the RL agents even traveled less distance?

3. Line 321 - 329, how to formulate the Q-value variance? Is it empirically computed or parametrically formulated?

4. Figure 6 top row and Figure 12 top row, what is the meaning of "distance changes" and the x-axis? What is the reason for the change to be linear?

5.Line 603, what is the evaluation criteria or quantification of "risk-awareness" for RL agents? It is ambiguous to claim that RL agents behavior should resemble mice's. Besides, how does risk-awareness further benefit the RL agents if the survival rate is not the primary focus?

**Other Comments Or Suggestions:**

No

**Other Strengths And Weaknesses:**

This paper provided an insightful work including both biological experiments and first-hand data for "predator-prey" to illustrate the risk-awareness and self-preservation, and provided feasible modifications to the previous method to make reinforcement learning model to capture similar patterns of mice's.

I have a few major concerns. The first one is about the motivation for including the risk-awareness in RL models, especially since the success rates between the mice and the simulation are similar (and the setup of the RL model in this paper is arguably designed to be "reckless", since the bonus and penalty are of the same scale). Second, "risk-awareness" is ambiguous and subjective, and it lacks quantification. The authors also need to provide more evidence to claim the benefits of incorporating it into the RL models.

**Questions For Authors:**

No

**Relation To Broader Scientific Literature:**

This paper proposed two mechanisms in the field of reinforcement learning to introduce risk-awareness instinct to the model. The first one is based on Post Traumatic Stress Buffer (PTSB) (Al Abed et al., 2020) and the second one is extension to Surprise Minimization in Reinforcement Learning
(SMiRL) (Berseth et al., 2019).

In 6. What about LLM agents, the paper discussed the performance of large language model under the same simulation setup. I am not convinced by the necessity of discussion about LLM which seems irrelevant to the previous parts about reinforcement learning.

**Theoretical Claims:**

Not applied

---

> ### Author Rebuttal · Authors · 2025-03-31
>
> We thank the reviewer for the thorough and valuable feedback!
>
> We would like to address several key points:
>
> **Q: Motivation for Risk-Awareness in RL**
>
> A:  Our paper aims not to improve RL performance but to understand differences between biological and artificial learning. This basic research is crucial for advancing both fields. Modern RL has diverged significantly from biological principles, and understanding these differences has scientific value regardless of performance metrics. Many AI breakthroughs (CNNs from visual cortex, attention mechanisms from human vision, and Hierarchical RL from human planning) originated from biological insights. We aim to understand how and why biological risk assessment strategies differ from current RL approaches, which serves important scientific goals beyond conventional performance evaluation.
>
>
> **Q: How to Quantify Risk Awareness?**
>
> A: We have provided multiple objective indicators in the paper, including waiting behavior (mice pausing to assess risks), diverse path selection and wall-following behavior (shown in visitation density comparisons), and more conservative movement patterns (changes in average distance before/after predator detection).
> To further quantify these behaviors, we conducted additional quantitative analyses:
> -  Waiting behavior: Defined as movement with distance change <0.1 units within the first six steps, our analysis of 1,000 trajectories shows standard TD-MPC exhibits 0% waiting behavior, while mice demonstrate 27.7% and our improved method shows 32.4%.
> -  Episode length: Standard RL agents complete tasks with an average length of 7.09 steps, while our improved method shows 13.89 steps and the mouse shows 14.04 steps.
>
> These quantitative measures demonstrate that our proposed mechanisms enable RL agents to exhibit risk perception abilities more similar to mice. We will include these results in the future.
>
>
>
> **Q : Could you clarify the meanings of 'receives +1 reward at the goal' and 'puffed by the predator,' and would increasing the penalty (beyond ±1) make RL agents more cautious compared to mice?**
>
> A: In our setup, the hexagonal environment has a starting point at one vertex and a goal point on another. The agent (prey) receives a reward of +1 only upon reaching the goal point, which mirrors the real mouse experiment where mice receive water as a reward at the goal. The agent receives a penalty of -1 when "puffed" by the predator, which occurs when their distance is below 0.1 units (27.5cm in real environment). This corresponds to the air puff stimulus used in the physical mouse experiments.
>
> The initial equal magnitude (+1/-1) for rewards and penalties was chosen to reflect that both water reward and air puff are relatively mild stimuli. We did experiment with larger penalty values (up to 200x). However, even with increased penalties, the RL agents still exhibited fundamental differences from mice in that they would not display waiting behavior at the start point. This behavioral difference highlights the need for additional mechanisms beyond simple reward scaling.
>
>
> **Q: How to formulate the Q-value variance?**
>
> A: We would like to highlight that this is explicitly detailed in our paper. The Q-value variance is empirically computed by uniformly sampling actions from the action space and calculating the variance of their Q-values (as stated on the right side of lines 279-284).
>
> **Q: Figure 6 top row and Figure 12 top row, what is the meaning of "distance changes" and the x-axis? What is the reason for the change to be linear?**
>
> A: The plots show how goal-agent distance changes before and after predator detection. (This information is in the caption, maybe too small now.) The linear trend emerges naturally because we are considering an average of 1000 episodes. This big number average makes mice’s and also RL’s trend to be linear.
>
> **Q: Why Discuss LLMs?**
>
> A: We included LLMs to compare how different agents behave in similar environments, tying back to the theme of "machines and mice." The results suggest risk-awareness is not just an RL issue but a broader challenge in AI cognition. The LLM experiments are supplementary, not a core focus.
>
> **Q: Some other clarification**
>
> A: The survival rate is the same as the success rate.
>
> In the end, we greatly appreciate the reviewer's insights and look forward to further discussions!

---

> > ### Comment · Reviewer_YZj9 · 2025-04-04
> >
> > Thank you for the clarifications! I would like to raise my score to "weak accept".

---

> > > ### Author Response · Authors · 2025-04-06
> > >
> > > Thank you so much for your supportive comment. If at your convenience, would you mind kindly updating the score in the system as well? We truly appreciate your time and support!

---

### Official Review · Reviewer_TtQ7 · 2025-03-13

**Overall Recommendation:** 4

**Summary:**

This paper compares the learned behaviors of biological mice and simulated RL agents in a predator-avoidance maze environment. The authors highlight the disparities in behavior and propose two mechanisms to bridge the gap between a TD-MPC2-trained simulated agent and biological mice. Finally, they compare the performance of an LLM-agent and show its similarity to an RL agent.

**Claims And Evidence:**

I'm mostly fine with the claims, but they are generalized across RL agents when, in fact, they may be an artifact of specific task specifications in the simulation and the choice of algorithms (discussed in the upcoming sections). That said, since the authors focus solely on comparing behaviors and do not claim biological plausibility, I suggest only minor rewrites.

**Essential References Not Discussed:**

The related work section primarily motivates RL as a useful framework for understanding biological intelligence. However, it does not discuss prior works similar to theirs or explain why RL alone is sufficient to generate the behaviors observed in mice.

Additionally, the section appears somewhat hastily written. For instance, a paragraph begins with the vague statement, "Fundamental challenges persist" (line 438). While the related work lists studies that explore the intersection of RL and neuroscience, it does not clearly connect them to the present study. For example, a more relevant discussion could address why model-based RL is specifically chosen to model rat behavior.

Please read the papers I mentioned in this review and consider citing them if relevant.

**Experimental Designs Or Analyses:**

**Task Specification**

I have some concerns about how the task is specified in simulation. I appreciate the effort the authors put into aligning the simulation with the real-world task. However, the disparity in qualitative behaviors may stem from the use of an episodic RL formulation with discounting and sparse rewards.

An RL agent taking the shortest path to the goal is likely an artifact of choosing discount factor $\gamma < 1$ with sparse rewards, as this setup inherently incentivizes shortest-path solutions. While I do not know the biological mechanisms underlying mouse intelligence, it seems highly unlikely that they align with an episodic RL framework with discounting, as used in these experiments.

To understand the impact of task specification in goal-reaching tasks, Vasan et al. (2024) is a useful reference. Their work demonstrates that three seemingly equivalent task specifications in RL can lead to different final performance outcomes, even when using the same learning algorithm.

**Comparisons with Other RL Methods **
- TD-MPC2 is a model-based RL algorithm. I am curious whether model-free algorithms such as PPO or SAC achieve similar final performance to TD-MPC2.

**LLM Agent Description**
- The action space of the LLM agent is unclear. Does it select an action at every timestep, or does it generate an entire trajectory at once? For a fair comparison with TD-MPC2, the LLM should be evaluated in a setting where it selects an action at each timestep.


**References**
- Vasan, G., Wang, Y., Shahriar, F., Bergstra, J., Jagersand, M., & Mahmood, A. R. (2024). Revisiting Sparse Rewards for Goal-Reaching Reinforcement Learning. Reinforcement Learning Journal, 4, 1841–1854.

**Methods And Evaluation Criteria:**

- In Section 3, the authors describe Deep Q Network (DQN) and Soft Actor-Critic (SAC), along with other behavior cloning and offline RL methods in Appendix C. However, all the plots in the main paper only use TD-MPC2. Why is this the case? If the results from the other methods are not used, perhaps the unnecessary text could be removed?

- The ideas of Post Traumatic Stress Buffer (PTSB) and Selective Memory Sampling are reminiscent of Prioritized Experience Replay (Schaul et al., 2015). Did the authors consider this approach? Would it be accurate to say that the proposed approach is one instantiation of prioritized experience replay?

- TD-MPC2 uses Model-Predictive Path Integral (MPPI) to sample actions. In simpler terms, it generates multiple rollouts of planning horizon $H$ and picks the action sequence that yields the highest predicted return, typically using a weighted averaging scheme based on the rollouts' performance. The default implementation of TD-MPC2 only considers a horizon length of 3 or 5, which is very small. What is the choice of of planning horizon $H$ in the experiments? This could potentially impact your results.

- The proposed Variance-Penalized Temporal Difference (TD) Learning aims to reduce risky behaviors by avoiding states with high uncertainty. However, it is unclear whether risk and uncertainty are inherently the same. For example, early in learning, Q-values may appear uncertain due to limited exploration, but this does not necessarily imply that those states are inherently risky.

**References**
- Schaul, T., Quan, J., Antonoglou, I., & Silver, D. (2015). Prioritized experience replay. arXiv preprint arXiv:1511.05952.

**Other Comments Or Suggestions:**

- I'm open to reconsidering my score if 1) the authors update the related work section to better position their study and share it during the rebuttal, and 2) address my concerns regarding the methods and evaluation criteria.


**Typos, etc**
- The citation in the very first line is incorrect. It is Sutton & Barto (2018). Not just Sutton (2018). In APA: "Sutton, R. S., & Barto, A. G. (2018). Reinforcement learning: An introduction. MIT press"

**Other Strengths And Weaknesses:**

**Strengths**
- The paper is well-written and enjoyable to read.
- The plots and illustrations are aesthetically pleasing and effectively highlight the behaviors the authors aim to showcase.
- The analysis is timely, given the increasing role of AI in society. Comparisons with biological intelligence are valuable and much appreciated.

**Weaknesses**
- The authors do not provide strong explanations or hypotheses for why RL agents behave differently from mice.
- The proposed mechanisms for promoting risk aversion feel ad hoc. I encourage the authors to consider additional factors, such as the choice of algorithm and task specification.

**Questions For Authors:**

- What is the planning horizon \( H \) used in your TD-MPC2 experiments?
- What is the chosen discount factor \( \gamma \)?
- Evolution optimizes for reproductive success, which inherently emphasizes self-preservation behaviors. In contrast, an RL agent lacks such pressure and is only optimized for success and failure. Do you think this could partly explain the differences in behavior? If so, could this be encoded through rewards?
- Figure 11: The authors state that the proposed “VP-TD-MPC2 + PTSB” achieves an 86.1% visitation overlap with a mouse. However, the trajectories still appear visually different. Do you have any hypotheses on why this gap remains?
- Would a "baiting" behavior emerge with longer training in the simulation?

**Relation To Broader Scientific Literature:**

This aligns with a broader body of work, including Vaxenburg et al. (2024) and Singh et al. (2023), which compare biological intelligence and RL agents, conducting behavioral analyses to assess biological plausibility and differences

**References**
- Vaxenburg, R., Siwanowicz, I., Merel, J., Robie, A. A., Morrow, C., Novati, G., ... & Turaga, S. C. (2024). Whole-body simulation of realistic fruit fly locomotion with deep reinforcement learning. bioRxiv, 2024-03.
- Singh, S. H., van Breugel, F., Rao, R. P., & Brunton, B. W. (2023). Emergent behaviour and neural dynamics in artificial agents tracking odour plumes. Nature machine intelligence, 5(1), 58-70.

**Theoretical Claims:**

N/A

---

> ### Author Rebuttal · Authors · 2025-04-01
>
> We sincerely appreciate the reviewer's thorough and valuable feedback. Below we address each point systematically:
>
> **Q: Regarding algorithm selection and comparison**
>
> A: We experimented with DQN and SAC alongside TD-MPC2 but focused on TD-MPC2 because: (1) Model-based approaches together with TD learning better align with biological decision-making (Daw et al., 2011), (2) DQN and SAC showed less stable convergence and inconsistent trajectories, and TD-MPC2 produced more stable behaviors.
>
> Notably, with sufficient training, all RL methods ultimately exhibited similar "reckless" behaviors (lacking waiting periods and showing minimal exploration), confirming that our observations about behavioral gaps are not specific to one algorithm.
>
>
> **Q: Concerning Prioritized Experience Replay (PER)**
>
> A: While our approach shares conceptual similarities with PER, our experiments showed standard prioritized replay failed to change the RL behavior. The key distinction is our specific amplification of negative experiences to model trauma-like responses, simulating high-risk encounters in a biologically-inspired manner, rather than prioritizing based solely on TD error magnitude.
>
>
> **Q: As for risk and uncertainty**
>
>
> A: We observe that while initial training phases do show high variance across the environment (reflecting exploration uncertainty), this pattern quickly evolves to specifically highlight areas with risk (predator encounters). After sufficient exploration, Q-variance remains low near walls and starting positions but stays consistently high in predator-dense regions, confirming our mechanism captures genuine risk rather than mere exploration uncertainty.
>
> **Q: On Task Specification, Evolutionary Pressure, and Implementation Details**
>
> A:  We agree that evolutionary pressure contributes to these behavioral differences. Our reward structure (-1/+1) mirrors our Cell Reports study paradigm (not cited due to double-blinding), where mice received mild water rewards and air puffs of similar intensity. Increasing penalty values failed to encourage cautious behavior like waiting behavior at the entrance. We chose episodic RL and sparse rewards specifically to maintain consistency with our mouse experiments.
>
> The 0.995 discount factor was selected to encourage longer-term planning that better matches mouse behavior, though it lacks direct biological explanation. Importantly, behavioral differences persisted across all tested discount factors (0.9-0.995).
>
> We used H=3 after experimenting with longer horizons (H=6,10), which degraded performance due to compounding prediction errors. When H is too big, the prey can even ignore the predator.
>
> No baiting behavior emerged even with extended training (up to 10M steps).
>
> The above shows fundamental risk assessment differences rather than training duration issues.
>
> **Clarification On LLM**
>
> A: Our LLM agent makes decisions at each timestep, identical to TD-MPC2, ensuring a fair comparison despite the computational cost.
>
>
> **We've improved the related work section**
>
> A:  Recent studies by Vaxenburg et al. (2024) and Singh et al. (2023) demonstrate the value of comparing biological and artificial behaviors in navigation tasks. Our choice of model-based RL draws from Daw et al. (2011), showing animals maintain internal models for decision-making, with striatal prediction errors paralleling TD learning. Regarding whether standard RL suffices for mouse-like behaviors, Mattar & Daw (2018) showed that prioritized memory access explains biological planning processes, inspiring our PTSB. Blanchard et al. (2011) documented neural risk assessment mechanisms in rodents aligning with our variance-penalized approach. These studies collectively suggest that while standard RL alone cannot reproduce mouse behaviors, our biologically-inspired modifications bridge this gap.
>
>
> **Regarding weaknesses**:
> Our proposed mechanisms are not ad hoc, they are directly motivated by observed mouse behavior, as detailed in our response to Reviewer GbX7.
>
>
> **REF**
>
> - Vaxenburg, R., Siwanowicz, I., Merel, J., Robie, A. A., Morrow, C., Novati, G., ... & Turaga, S. C. (2024). Whole-body simulation of realistic fruit fly locomotion with deep reinforcement learning. bioRxiv, 2024-03.
> - Singh, S. H., van Breugel, F., Rao, R. P., & Brunton, B. W. (2023). Emergent behaviour and neural dynamics in artificial agents tracking odour plumes. Nature machine intelligence, 5(1), 58-70.
> - Daw, N.D., et al. "Model-based influences on humans' choices and striatal prediction errors." Neuron 69.6 (2011): 1204-1215.
> - Mattar, M.G., and Daw, N.D. "Prioritized memory access explains planning and hippocampal replay." Nature Neuroscience 21.11 (2018): 1609-1617.
> - Blanchard, D.C., et al. "Risk assessment as an evolved threat detection and analysis process." Neuroscience & Biobehavioral Reviews 35.4 (2011): 991-998.
>
> In the end, we thank the reviewer again for insightful comments! We look forward to further discussions with you.

---

> > ### Comment · Reviewer_TtQ7 · 2025-04-04
> >
> > I thank the authors for their detailed response, which clarifies several of the questions I had earlier. I also appreciate the updated related work paragraph. I want to emphasize that I like this line of work and would like to help make it clearer and more compelling to the broader RL community. At this point, I will stick with my current score. However, if the authors can provide a reasonable plan to address my remaining questions and concerns in the final draft, I would be happy to raise my score.
> >
> > ---
> >
> > **> Mismatch between generality of claims and tailored solutions**
> > I believe some of the criticism raised by other reviewers stems from the mismatch between the generality of the paper’s claims and the bespoke nature of the proposed mechanisms. For example, both the Post-Traumatic Stress Buffer and the variance-penalized TD update appear to be carefully crafted solutions aimed at inducing risk-averse behavior similar to that seen in biological rats. Since the claims made in the paper pertain broadly to biological and artificial agents, it's worth examining whether the proposed mechanisms are actually general or if they are domain-specific solutions.
> >
> > ---
> >
> > **> Horizon length of 3 with TD-MPC**
> > As you can imagine, a planning horizon of 3 with TD-MPC is quite short—equivalent to only a few hundred milliseconds in real-world time. I suspect that a more capable model-based RL method, one that supports longer-horizon planning, could achieve risk-averse behavior purely through decision-time planning. Would the authors agree with this? If so, should we view the proposed mechanism as a stopgap solution in lieu of stronger planning methods?
> >
> > ---
> >
> > **> Comparison with Prioritized Experience Replay (PER)**
> > The authors mention that, while their approach shares conceptual similarities with PER, standard prioritized replay did not alter the agent's behavior in the same way. The key distinction, as stated, is that their method amplifies negative experiences to simulate trauma-like responses, as opposed to simply prioritizing transitions based on TD error.
> >
> > However, I couldn't find any mention of PER in the submitted draft. If you have experimental results comparing your approach with PER, please include them in the paper. Given the relevance of PER, it’s important to cite it directly and explain its limitations in your context, along with the motivation for your proposed mechanism.
> >
> > ---
> >
> > **> Figure 11 – Visitation overlap**
> > The paper states that “VP-TD-MPC2 + PTSB” achieves an 86.1% visitation overlap with mouse trajectories. However, the qualitative difference between the trajectories remains noticeable. Do you have any hypotheses as to why this gap persists? This point is still unclear to me and would benefit from further explanation.
> >
> > I look forward to your response!

---

> > > ### Author Response · Authors · 2025-04-06
> > >
> > > We sincerely thank your thoughtful feedback, which significantly helps us clarify and strengthen our paper!
> > >
> > > ---
> > >
> > > **Q: Generality of Claims vs Specific Solutions**
> > >
> > > A: Our work indeed focuses on a specific predator-prey domain, and the proposed mechanisms were intentionally designed within this context. However, predator-prey dynamics represent fundamental survival interactions observed across both natural and artificial systems, as highlighted by Tsutsui et al. (2024) and Marrow et al. (1996). These studies emphasize such scenarios as key paradigms for investigating adaptive behavior.
> > >
> > > In the final draft, we will refine our claims to acknowledge the domain-specific nature of our mechanisms. While our approach is tailored to predator-prey settings, we believe it offers valuable insights into computational models of biological caution. We view these dynamics as a principled and biologically grounded testbed for studying adaptive behavior under threat, which we hope makes our findings more relevant to the RL community.
> > >
> > > **Ref**
> > >
> > > - Tsutsui, Kazushi, et al. "Collaborative hunting in artificial agents with deep reinforcement learning." Elife 13 (2024): e85694.
> > >
> > > - Marrow, Paul, Ulf Dieckmann, and Richard Law. "Evolutionary dynamics of predator-prey systems: an ecological perspective." Journal of mathematical biology 34 (1996): 556-578.
> > >
> > > ---
> > >
> > > **Q: Horizon length of 3**
> > >
> > > A: The horizon length of 3 is well-suited to our experiments. Our task mirrors real mouse behavior: excluding waiting periods, mice need 2-3 seconds to reach the goal. Given our simulation timestep (0.25 seconds per action), successful task completion in RL requires 8-12 steps.
> > >
> > > Why 0.25s? This aligns with findings showing that rodents make approximately 2-5 decisions per second during navigation (Resulaj et al., 2009). Through environment setup, we determined that the 0.25s provides an optimal balance between biology realism and learning performance.
> > >
> > > To further investigate this issue, we conducted additional experiments using Dreamer-v3, which employs an LSTM module to handle longer horizons. Even with a horizon of 15, Dreamer-v3 failed to replicate cautious behaviors observed in mice. This indicates behavioral differences arise not from horizon limitations, but from principles addressed by our biologically-inspired mechanisms.
> > >
> > > We will expand on these experimental design choices and their rationale.
> > >
> > > **Ref**
> > >
> > > - Resulaj, A., Kiani, R., Wolpert, D.M., & Shadlen, M.N. (2009). Changes of mind in decision-making. Nature, 461(7261), 263-266.
> > >
> > > ---
> > >
> > > **Q:Comparison with Prioritized Experience Replay (PER)**
> > >
> > > Our results show that PER offers partial benefits but does not fully address the challenges in our scenario. Specifically, PER achieves an 80% survival rate within 40,000–50,000 steps (vs 60,000–70,000 with standard replay). However, density plots indicate minimal behavioral differences after training. Thus, PER improves training efficiency but has limited impact on the behaviors. This limitation motivated our proposed mechanism, which explicitly amplifies negative experiences simliar to biological trauma responses.
> > >
> > > We will explicitly cite PER, compare with it, and clarify our design motivation in the final manuscript, supported by experimental plots.
> > >
> > > ---
> > >
> > > **Q:  Fig 11 and why this gap persists**
> > >
> > > A: The reported 86.1% visitation overlap indicates that the RL agent explores approximately 86% of the same spatial area as the mouse across all episodes. This metric reflects similarity in exploratory coverage (whether both agents have been to the same states), rather than a one-to-one match in how frequently or in what manner they visit those states.
> > >
> > > It is also true that in Fig 11, the final converged policies still display qualitative differences. Mice exhibit stronger wall-following behavior, while our improved RL agents tend to use central regions more.
> > >
> > > Our hypothesis:
> > >
> > > - This behavioral difference stems from the structure of the RL state space. Having thoroughly explored the outer wall during training, the agent doesn't need to maintain constant wall contact for safety. Upon reaching the arena's top, it can confidently consider the task solved.
> > >
> > > - The original TDMPC2 agent exclusively used central regions. While our improved RL agent now captures some cautious mouse-like behaviors, it has learned that central areas provide more efficient paths to the goal when safety is assured. In contrast, mice persist in wall-following regardless of task mastery, likely due to hardwired behavioral priors.
> > >
> > > We acknowledge this distinction was previously unclear and will revise the text to emphasize that visitation overlap does not imply similarity in behavioral patterns or density plot distributions.
> > >
> > > ---
> > >
> > > We hope these clarifications and planned revisions address your concerns. We are very grateful for your detailed feedback and your openness to reconsidering your score!
> > >
> > > Your suggestions have already helped us strengthen the work in several ways!

---

### Official Review · Reviewer_bEMR · 2025-03-14

**Overall Recommendation:** 3

**Summary:**

This paper investigates the behavioral differences between biological mice and reinforcement learning (RL) agents in a predator-avoidance maze environment. The authors find that RL agents lack preservation instincts, often taking risky, efficiency-driven paths without assessing potential threats, in contrast to biological mice. To address this discrepancy, the paper introduces two novel mechanisms designed to encourage biologically inspired risk-avoidance behaviors in RL agents: (i) Post Traumatic Stress Buffer (PTSB), which mimics biological trauma responses, and (ii) Variance-Penalized Temporal Difference (TD) Learning, which integrates action uncertainty by penalizing high Q-value variance and promotes more risk-averse decision-making. Experimental results demonstrate the effectiveness of both proposed mechanisms. Additionally, the authors introduce ChatGPT-4 to explore the behavior of large language model (LLM) agents in a controlled environment. However, experiments reveal that LLM-based agents exhibit risk-taking tendencies similar to those of RL models, suggesting that achieving biological alignment remains a broader challenge in AI.

## update after rebuttal
I keep my initial rating.

**Claims And Evidence:**

Good.

**Essential References Not Discussed:**

none.

**Experimental Designs Or Analyses:**

Good.

**Methods And Evaluation Criteria:**

Good.

**Other Comments Or Suggestions:**

No.

**Other Strengths And Weaknesses:**

Strengths:

1. The paper presents a systematic and quantitative analysis of the behaviors of mice compared to reinforcement learning (RL) agents in a predator-prey maze, revealing significant differences in risk assessment and exploratory strategies.

2. The writing of this paper is good. The proposed PTSB and Variance-Penalized TD learning designs are well-motivated.

3. The proposed novel methods for altering RL behaviors to align with biological processes are effective and have potential application value in life science research and other related fields.

Weaknesses：

1. The experiments are conducted in Cellworld Gymnasium, raising questions about whether the proposed mechanisms would generalize to other, more diverse environments or tasks. Could the authors provide more analysis or change the environment to demonstrate the generalization of the mechanisms?

2. This paper utilizes ChatGPT-4 to conduct the experiments. However, the analysis appears limited, partly due to the reduced number of trial runs, which may lead to unreliable conclusions.

**Questions For Authors:**

No

**Relation To Broader Scientific Literature:**

None.

**Theoretical Claims:**

None.

---

> ### Author Rebuttal · Authors · 2025-04-01
>
> We appreciate the reviewer's thoughtful feedback. Regarding the two main concerns raised:
>
> **Q: On generalization across environments: The experiments are conducted in Cellworld Gymnasium, raising questions about whether the proposed mechanisms would generalize to other, more diverse environments or tasks. Could the authors provide more analysis or change the environment to demonstrate the generalization of the mechanisms?**
>
> A: We acknowledge that generalization is an important consideration. Our research primarily aims to understand fundamental behavioral differences between biological and artificial agents in predator-avoidance scenarios, rather than demonstrating broad environmental generalization. We focused on a controlled environment that precisely mirrors our biological experiments, allowing for direct behavioral comparisons.
>
> While broader generalization testing is valuable for future work, our current focused comparison provides crucial insights into the computational principles underlying biological risk assessment. These principles (memory amplification of negative experiences and uncertainty aversion) are likely to transfer to other risk-assessment scenarios, though the specific behavioral manifestations may vary with environment. Future work will examine how these mechanisms generalize across different environmental structures and predator behaviors.
>
> We truly appreciate your recognition of our systematic quantitative evaluation approach and the biological motivation behind our proposed mechanisms. Our future work will indeed explore applications in more complex environments, particularly in AI safety domains and autonomous navigation systems where risk assessment is critical.
>
>
> **Q: Regarding the ChatGPT-4o analysis: This paper utilizes ChatGPT-4o to conduct the experiments. However, the analysis appears limited, partly due to the reduced number of trial runs, which may lead to unreliable conclusions.**
>
> A: While we conducted approximately 100 trajectories (limited by computational constraints of running GPT-4o), our experimental design ensures comprehensive coverage through systematic variation of predator spawn locations across all relevant scenarios. The remarkable consistency of ChatGPT-4o's behavior across these diverse configurations provides strong evidence for our conclusions. Our analysis reveals that regardless of predator positioning, ChatGPT-4o consistently adopts strategies similar to baseline TDMPC2, reinforcing our findings about the fundamental behavioral gap between artificial and biological agents. We will include additional visualizations showing predator spawn distributions to further support this conclusion.
>
>
>
> We look forward to further discussions with you!

---

### Official Review · Reviewer_GbX7 · 2025-03-14

**Overall Recommendation:** 2

**Summary:**

The authors devise several RL agents for a navigation task that involves reaching a goal while avoiding a predator. The task is closely modeled after an actual biological experiment with real mice. Based on the observation of systematic behavioral differences between biological and artificial agents in this predator-prey environment, they suggest two additions to a hybrid RL algorithm, which combines TD learning and MPC involving a deep encoder-decoder model: 1) a replay buffer mechanism that resamples episodes with close encounters with the predator during training more often and 2) a penalty term in the TD error that penalizes reward with the variance of the Q-values across actions in that state. Empirical results show that these two modifications result in an increased overlap of cell visitations between the simulated agent and mice.

**Claims And Evidence:**

The claims are that
1) survival performance,
2) state visitation distributions,
3) wall following behavior,
4) initial waiting
are closer to mice's behavior in the proposed RL model. 1-2 are evaluated quantitatively, and 3-4 are evaluated qualitatively. However, while 1 does not seem to differ (a statistical test may help here), the used measure for 2 shows an improvement, but the typical wall-following trajectories of mice are still not reproduced.

**Essential References Not Discussed:**

Sorry, not sure what I would pick out here as "essential".

**Experimental Designs Or Analyses:**

The specific behavior of the RL agent contrasts with the behavior of the mice. However, many choices in setting up the RL agent have not been further discussed or investigated. First, the reward for encountering the predator is fixed. Second, the agent does not explore the. maze but encouraging exploration e.g. by adjusting \beta in the third equation on page 3. Third, the environment is clearly partially observable, which is not explicitly modeled. Fourth, the predator is always detected, no matter in what direction it is relative to the agent, including being in its back, which does not correspond to the mice’s field of view. Fifth, the conclusions are drawn based on predator behavior determined by the parameters of algorithm 1, and more aggressive attack behavior may suffice to obtain more

“Experiments (Figure 7) indicate that 50% (ranging from 0% to 100% in increments of 25%) is the optimal percentage for this buffer.” How generalizable is this result, or how much does this pertain to the specifics of the current setup?

While the authors report that “TDMPC-2 agent has a 20.9% overlap with mice, while VP-TDMPC-2 achieves 86.1%”, looking at figure 11 seems to suggest very different behavior and still limited wall-following behavior.

**Methods And Evaluation Criteria:**

The algorithmic choices, such as the replay buffer and altered cost function are ad hoc, and at ICML it would be great to see a more in-depth motivation and analysis of these choices.

It would be very helpful to find more principled and quantitative measures of similarity between trajectories and between strategies of behavior in the predator-prey environment to better quantify and understand the similarities and differences in behavior between mice and models.

**Other Comments Or Suggestions:**

Characterizing DQN as online RL is a bit difficult, given that it uses the replay buffer, which makes it more of an offline method.

The color mapping is wrong in the caption of figure 3.

**Other Strengths And Weaknesses:**

This is an interesting and important direction for neuroscience and ethology, but I think that the paper in its current form is not at the core of the interests of the ICML community but would elicit more excitement at a conference such as From Animals to Animats.

**Questions For Authors:**

How was “Memory Amplification” implemented?

Where do the encounters with the predator happen? A density plot of the encounters with the predator would help understand the changes in agent behavior given the modified algorithms.

Would different rewards in the problem's setup, especially for being caught by the predator, change behavior?

Are the differences in behavior mostly attributable to not using a formulation of partial observability?

**Relation To Broader Scientific Literature:**

There is a wealth of literature on modeling the behavior of animals, particularly mice with RL. The literature on replay buffers since DYNA is extensive, similarly the literature on implementing forms of risk aversion in RL.

I would be very careful with naming an algorithm as the present one, “Post Traumatic Stress Buffer (PTSB),” just out of respect for the medical profession and the patients suffering from such conditions. Maybe there is a way to reformulate this.

**Theoretical Claims:**

There are no theoretical claims involving theorems.

---

> ### Author Rebuttal · Authors · 2025-04-01
>
> We appreciate the reviewer's thorough feedback and would like to address several key points:
>
> **Q: Why PTSB and VP-TDMPC2**
>
> A:  Our PTSB models rodents' responses to adverse stimuli. Even with mild aversive stimuli (air puffs), mice show pronounced fear responses after few exposures, exhibiting PTSD-like memory amplification. LeDoux (2000) demonstrates how rodents overweight negative experiences during risk assessment, which we implement by selectively amplifying memories of predator encounters.
>
> Our Variance-Penalized approach stems from observing mice frequently waiting at the starting position before making decisions - behavior suggesting internal risk evaluation. Rushworth & Behrens (2008) found mammalian brains specifically encode uncertainty in decision-making circuits, with orbitofrontal cortex activation correlating with outcome uncertainty. Behaviorally, animals consistently avoid high-variance options even with identical expected values. We evaluated multiple uncertainty measures before selecting variance as most biologically plausible and computationally effective.
>
> **REF**
>
> - LeDoux, Joseph E. "Emotion circuits in the brain." Fear and anxiety (2013): 259-288.
> - Rushworth, Matthew FS, and Timothy EJ Behrens. "Choice, uncertainty and value in prefrontal and cingulate cortex." Nature neuroscience 11.4 (2008): 389-397.
>
> **Q: Predator Encounter Density and Behavioral Implications**
>
> A: Predator encounters primarily occur in central maze regions, as our density plots show. This distribution explains why our mechanisms produce mouse-like behaviors: when combined with PTSB (amplifying learning from central encounters) and variance-penalized training, agents develop:
>
> -  increased waiting at the starting position to assess risk
> -  stronger preference for wall-following paths that avoid these high-risk central areas
>
> We will add density plots to illustrate this relationship between encounter locations and resulting behavioral adaptations.
>
>
>
> **Q: Memory Amplification Implementation and PTSB Generalizability**
>
> A: In standard replay buffers, negative experiences typically constitute only about 10% of collected data. Our approach ensures higher representation (50%) in each training batch. This ratio showed optimal training efficiency in our experiments and the principle generalizes to other algorithms like DQN and SAC, though optimal ratios may vary.
>
>
> **Q: Environmental Design Choices and Partial Observability Effects**
>
> A: Regarding predator visibility and observation design: Mice have approximately 270° natural vision, and our experiments documented frequent head-turning behavior that gives them effective near-360° awareness. Our implementation matches this natural capability rather than imposing artificial constraints.
>
> We tested various observation spaces, including more limited observations (only prey/predator positions when visible), field-of-view restrictions, and observations without goal information. Importantly, all variations produced similar behavioral patterns—RL agents consistently took direct paths without waiting or wall-following behaviors. This consistency suggests the behavioral differences aren't primarily attributable to partial observability.
>
> Our current observation space specifically parallels actual mouse experimental conditions while maintaining computational tractability. For the robot, the same algorithm was used in both physical trials and simulation, enabling direct comparison between RL agents and mice under identical conditions.
>
> **Q: Wall-Following Behavior Limitations**
>
> A: We acknowledge certain limitations. Extreme wall-following remains challenging even when giving the agent extra rewards for being near the wall, particularly with predator presence. However, our approach achieves significant improvements in risk-aware behavior compared to baselines.
>
> **Q: Impact of Varying Rewards and Risk Quantification**
>
> A: The same question is asked by reviewer yzj9, and we have provided detailed answers addressing both reward variation effects and our risk quantification methodology in that response.
>
> **Q: DQN Classification and PTSB Terminology**
>
> A: We agree that RL terminology can be confusing. DQN is off-policy (learns from experiences generated by different policies) but is considered online in standard RL taxonomy because it continuously collects new data through environment interaction. We agree with the reviewer regarding the terminology confusion and will revise accordingly in the final version.
>
> **Q: ICML relevance**
>
> A: ICML welcomes interdisciplinary research. Biological systems have developed sophisticated risk-assessment mechanisms that could inform machine learning approaches in high-stakes scenarios. This research direction aligns with ICML's mission to advance AI capabilities through cross-disciplinary insights.
>
> In the end, we again appreciate the reviewer's insights and look forward to further discussions!

---

### Decision · Program_Chairs · 2025-05-01

**Decision:**

Accept (poster)

**Comment:**

This paper compares mice and reinforcement learning (RL) agents on specific behavioral measures in a navigation task where subjects must reach a goal while avoiding predators. The authors have done an excellent job modeling the task to closely match the one used in mouse experiments. Reviewers raised concerns regarding the generality of claims about biological and artificial agents, and suggested more systematic comparisons with a wider range of model-based and model-free RL agents. While this paper falls somewhat outside typical ICML submissions, I believe it represents an important research direction in aligning artificial and biological agents. Beyond baseline RL models, the authors also replicate the experiments using ChatGPT-4 as an example of a large language model (LLM). Although the paper and the LLM experiment could be more comprehensive in their comparisons and could better explain the motivations behind their methodological choices, I still consider this an interesting contribution to ICML. Therefore, I recommend weak acceptance, with the expectation that the authors will address the reviewers' comments (specifically from reviewers GbX7 and TtQ7) in the revised manuscript.